# CogMoE: Signal-Quality–Guided Multimodal MoE for Cognitive Load Prediction

**Aamir Bader Shah**[*,1]**, Yu Wen**[*,2]**, Renjie Hu**[1]**, Jiefu Chen**[1]**, Jose L Contreras-Vidal**[1]**, Xuqing Wu**[1]**, Xin Fu**[1,†]
[1]University of Houston    [2]Stony Brook University

## Abstract

The poor and variable quality of physiological signals fundamentally constrains reliable cognitive load (CL) prediction in real-world settings. In safety-critical tasks such as driving, degraded signal quality can severely compromise prediction accuracy, limiting the deployment of existing models outside controlled lab conditions. To address this challenge, we propose CogMoE, a signal-quality–guided Mixture-of-Experts (MoE) framework that dynamically adapts to heterogeneous and noisy inputs. CogMoE replaces conventional modality-based fusion with a quality-aware gating mechanism that integrates EEG, ECG, EDA, and gaze according to their estimated signal quality, shifting the basis of multimodal modeling from modality identity to signal quality. The framework operates in two stages: (1) quality-aware multimodal synchronization and recovery to mitigate artifacts, temporal misalignment, and missing data, and (2) signal-quality-specific expert modeling via a cross-modal MoE transformer that regulates information flow based on signal quality. To further improve stability, we introduce CORTEX Loss, which balances task accuracy, quality-aware representation refinement and expert utilization under noise. Experiments on CL-Drive and ADABase demonstrate that CogMoE outperforms strong baselines across all modality combinations and sequence lengths, consistently delivering improvements across diverse signal-quality conditions. Our code is publicly available at `https://github.com/shahaamirbader/CogMoE`.

## 1 Introduction

Accurate prediction of cognitive load (CL) is critical in safety-critical domains such as driving, aviation, and healthcare, where elevated mental demand degrades decision-making and reaction time. Recent advances in multimodal sensing (EEG, ECG, EDA, and gaze) have made large-scale CL prediction feasible. However, the fundamental bottleneck is not the lack of sensors or models, but the variable quality of physiological signals. In real-world conditions, these signals are often noisy, misaligned, or partially missing as a result of motion artifacts, electrode drift, sensor dropout, and other well-documented physiological noise sources (Giangrande et al., 2024; Anwer et al., 2024). Unlike controlled laboratory studies, practical deployments must cope with heterogeneous, unstable input streams, where a single corrupted modality can severely compromise prediction accuracy. Moreover, unlike traditional multimodal setups where modalities provide complementary information, EEG, ECG, EDA, and gaze in CL prediction largely reflect redundant views of the same underlying cognitive process (Martínez Vásquez et al., 2023). Thus, signal quality, rather than sensor availability or model capacity, is the true limiting factor for accurate and reliable CL prediction.

Existing approaches to CL prediction have largely focused on improving classification accuracy through single-modality modeling or naïve multimodal fusion (Angkan et al., 2024; Islam et al., 2020). Traditional machine learning methods treat signals independently, while recent transformer-based models have demonstrated cross-modal integration. Yet, two critical limitations remain. On the data side, many approaches assume clean inputs, neglecting the pre-processing and recovery needed to handle artifacts, temporal inconsistencies, and missing segments. On the model side,

---

[*]Equal contribution.
[†]Correspondence: Xin Fu (xfu8@central.uh.edu).

current frameworks typically assign experts by modality, lacking adaptive mechanisms to account for real-time variations in signal quality. As a result, their performance degrades severely in noisy or heterogeneous environments, with typical accuracy capped at 70–80% (Pulver et al., 2023; Angkan et al., 2024; Azizi & BabaAli, 2024).

Motivated by this observation, we introduce CogMoE, a signal-quality–guided Mixture-of-Experts (MoE) framework that reframes multimodal fusion around estimated signal quality rather than modality identity. CogMoE couples this quality-centric formulation with an end-to-end architecture: experts specialized for high-fidelity, noisy, and recovered signals, and a quality-aware gating mechanism that routes representations in real time based on estimated signal quality. Our framework integrates (1) Quality-Aware multimodal Synchronization and Recovery, which aligns signals across sampling rates and recovers missing data, (2) Signal-Quality–Specific Expert Modeling, which employs a cross-modal transformer and Dynamic Pathway Gating (DPG) mechanism for quality-aware routing, and (3) CORTEX Loss, which balances task accuracy, quality-aware representation refinement and expert utilization to maintain consistent performance under noisy and incomplete signals. Together, these components form an end-to-end, adaptive, quality-aware design that directly addresses the core bottleneck of CL prediction under heterogeneous sensor conditions. Extensive experiments on CL-Drive and ADABase confirm that CogMoE achieves state-of-the-art performance, outperforming strong baselines by up to 13 percentage points, while maintaining stability under varying signal quality. Our key contributions are as follows:

  a) We develop a Quality-Aware Multimodal Synchronization and Recovery stage that aligns heterogeneous physiological signals and restores missing regions, providing a reliable foundation for subsequent quality-specific expert modeling.

  b) We introduce a signal quality-guided MoE that shifts the routing criterion from modality identity to estimated signal quality, with three experts tailored to distinct quality regimes: a High Fidelity expert for clean signals, a Noise Resilient expert for noisy inputs, and a Contextual Refinement expert for masked or recovered regions. A Dynamic Pathway Gating mechanism routes representations based on real-time signal quality estimates.

  c) We propose the CORTEX Loss, a set of quality-aware objectives designed to maintain task accuracy, support representation refinement, and prevent expert collapse. Its adaptive weighting schedule further stabilizes training under variable signal conditions, and ablations show that each component is required for reliable performance.

  d) Extensive experiments on CL-Drive and ADABase, covering all modality combinations and multiple sequence lengths, demonstrate that CogMoE achieves up to 13% and 9.5% accuracy gains over strong baselines, with consistent improvements in F1 across settings.

## 2    RELATED WORKS

### 2.1    COGNITIVE LOAD PREDICTION

CL prediction is essential for applications like human-computer interaction and healthcare. Early efforts focused on unimodal signals like EEG, which, despite its high temporal resolution, is highly susceptible to noise and lacks multimodal context (Pulver et al., 2023; Liu et al., 2024b; Yedukondalu et al., 2025). Similarly, ECG-based heart rate variability studies provide physiological insights but fail to capture the multidimensional nature of CL (Sriranga et al., 2023; Wang et al., 2024; Shi et al., 2024). Emerging multimodal approaches integrate EEG and ECG to enhance predictive accuracy (Azizi & BabaAli, 2024; Arts & Van den Broek, 2022), yet often overlook the complementary role of EDA and gaze data, which are crucial for comprehensive CL modeling. The recently released CL-Drive dataset (Angkan et al., 2024) provides a benchmark for CL classification across EEG, ECG, EDA, and gaze, advancing research in this area. ADABase (Oppelt et al., 2022) provides ECG, EDA, along with other signals in a simulated driving setting, and is used as a benchmark for multimodal cognitive load prediction.

Unlike previous works, we propose a quality-aware multimodal framework that first addresses the alignment of signals, then enhances data quality through recovery techniques, and finally integrates four physiological signals via a cross-attention transformer. This design directly targets the challenges of noise, misalignment, and missing data, thereby enabling consistent and reliable CL prediction under heterogeneous signal quality.

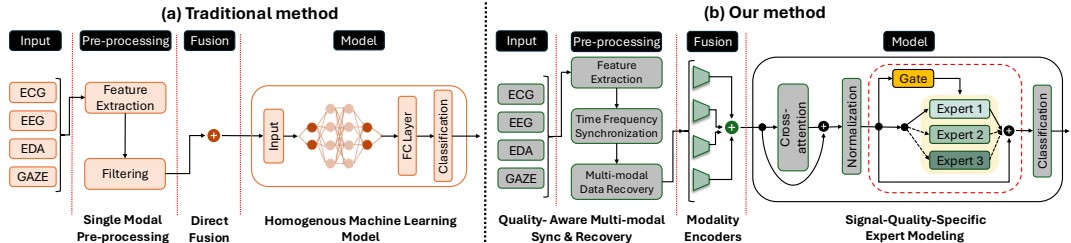

Figure 1: Comparison of (a) traditional method vs. (b) ours CogMoE.

## 2.2 MIXTURE OF EXPERTS NETWORKS

MoE (Mixture of Experts) architectures are gaining popularity for enhancing multimodal fusion by dynamically allocating computational resources to specialized experts for specific tasks. Sparse gating mechanisms, such as those introduced in (Shazeer et al., 2017), allow MoE models to activate expert networks selectively, improving efficiency and scalability. Recent advances in MoE models have addressed multimodal challenges such as missing modalities and high computational costs. For example, FuseMoE (Han et al., 2024) combines sparsely gated MoE layers with a Laplace gating function to boost robustness in image-text tasks, while FlexMoE (Yun et al., 2024) uses a missing modality bank and specialized routing strategies to handle diverse modality combinations, showing effectiveness in healthcare datasets like ADNI and MIMIC-IV. SwitchTransformer (Fedus et al., 2022) incorporates MoE layers within transformer blocks, reducing computational overhead by activating only a subset of experts per input. More recently, DeepSeek-V2 (Liu et al., 2024a) optimized MoE routing mechanisms to achieve state-of-the-art efficiency in large language models (LLMs).

Despite MoE's success in multimodal tasks, existing designs generally route experts based on semantic content or modality identity, implicitly assuming that input signals are reliable. This assumption does not hold in cognitive load prediction, where EEG, ECG, EDA, and gaze signals are often noisy, partially missing, or misaligned. Our work introduces a quality-guided MoE, in which experts are specialized for high-fidelity, noisy, and masked or recovered signals, and a dynamic gating mechanism that adaptively routes inputs according to signal quality. This shift from modality-based to quality-based expert assignment represents a fundamental reconfiguration of MoE, enabling adaptive modeling of physiological data under heterogeneous conditions.

## 3 METHODOLOGY

To achieve accurate CL prediction, we propose **CogMoE**, a signal quality–guided MoE framework designed to handle the noise, artifacts, and missing data that are inherent in physiological signals. Unlike the traditional methods, as illustrated in Figure 1, CogMoE operates in two complementary stages: first, a *quality-aware multimodal synchronization and recovery stage* acts as a pre-reconstruction step, aligning signals with different sampling rates and partially restoring corrupted or missing inputs; then, a *signal-quality-specific expert modeling stage* employs a cross-modal MoE transformer with dynamically gated experts specialized for high-fidelity, noisy, and missing signals, ensuring that information flow is regulated according to signal quality. Beyond these core components, CogMoE flexibly integrates physiological modalities, enabling effective prediction even when some are unavailable or degraded. Together, these stages allow CogMoE to deliver accurate and consistent CL predictions across diverse sensor conditions, enabling reliable multimodal inference under heterogeneous signal quality.

### 3.1 QUALITY-AWARE MULTIMODAL SYNCHRONIZATION AND RECOVERY

Physiological signals are often sampled at different rates and suffer from misalignment or missing segments. This motivates a two-step process, as shown in Figure 2: *Time-Frequency Synchronization*, which aligns signals across modalities, and *multimodal Data Recovery*, which restores incom-

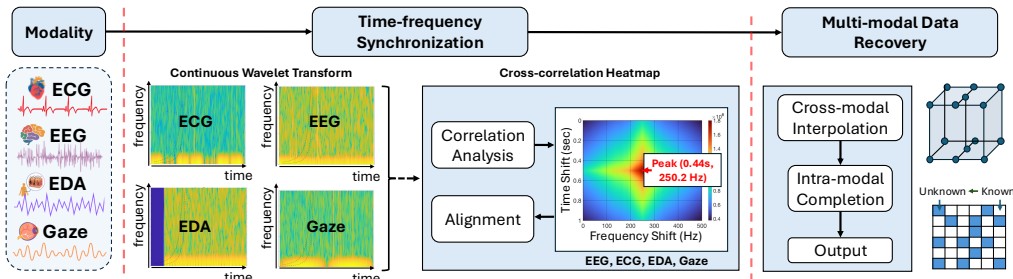

Figure 2: Overview of Quality-Aware Multimodal Synchronization and Recovery.

plete inputs through cross-modal recovery by leveraging intact modalities to fill gaps and intra-modal recovery to ensure within-modality consistency, providing cleaner inputs for downstream modeling.

### 3.1.1 TIME-FREQUENCY SYNCHRONIZATION

Misalignment across modalities such as EEG, ECG, EDA, and gaze corrupts cross-modal relations and degrades feature extraction. Standard methods like Dynamic Time Warping (DTW) are highly sensitive to spikes and rapid changes, often producing ambiguous or unstable alignments (Su et al., 2021; Doan et al., 2024). To address this, we adopt a synchronization strategy based on the Continuous Wavelet Transform (CWT), which projects signals onto a near-continuous time–frequency grid. This representation captures both transient and slow components, enabling more robust and interpretable alignment across heterogeneous physiological signals (Arts & Van den Broek, 2022).

Specifically, we first apply the CWT to each modality's signal $S_m$, converting the time-series data into a 2D time–frequency representation $W_m$. Using complex Morlet wavelets, this representation captures both high-frequency events (e.g., spikes) and low-frequency trends (e.g., slow ECG or EDA variations), thereby preserving transient as well as smooth components critical for alignment. To align signals across modalities, we cast the problem as 2D cross-correlation in time–frequency space. Compared to time-domain correlation, the time–frequency representation leverages spectral information, making alignments more stable under noise and non-stationary dynamics. The optimal alignment is given by the location of the maximum correlation; if multiple maxima exist, the alignment with the smallest temporal and frequency shifts is selected:

$$\Delta t^*, \Delta f^* = \arg\min_{t',f'} \left( \arg\max_{t',f'}(W_i * W_j)(t', f') \right) \tag{1}$$

where $\Delta t^*$ and $\Delta f^*$ denote the temporal and frequency shifts, respectively. These shifts are then applied to synchronize modalities. In practice, CWT is first applied channel-wise, and the aligned representations are aggregated into a modality-level tensor before entering modality-specific encoders, where lightweight channel attention or linear projections preserve intra-channel distinctions while producing compact embeddings. This process mitigates the ambiguities inherent in peak-matching methods, yielding more reliable synchronization for downstream modeling.

### 3.1.2 MULTIMODAL DATA RECOVERY

Sensor noise and dropout frequently create missing segments in multimodal physiological signals, which undermines reliable feature extraction and downstream modeling. To address this, we propose a two-step recovery strategy. First, cross-modal recovery uses intact modalities to infer whether and when key events occur. Second, intra-modal recovery restores modality-specific continuity, enforcing smooth transitions and periodic trends. Combined, these steps reconstruct missing regions consistently across modalities while preserving each signal's internal structure. Concretely, we first generate a binary mask $\mathcal{M}_m(t, f)$ for each modality by examining the aligned time–frequency representations of the other modalities. For each location $(t, f)$, we compute the aggregated energy $H_m(t, f)$ within a local neighborhood by summing contributions from all other modalities, and mark the region as missing if $H_m(t, f)$ exceeds a predefined threshold.

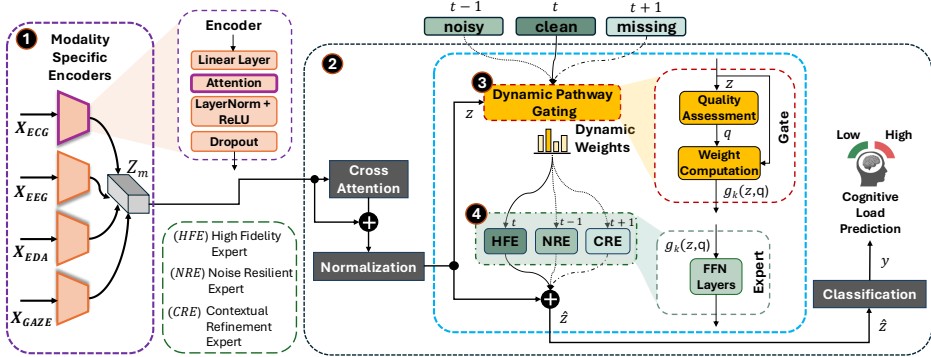

Figure 3: Overview of Signal-Quality-Specific Expert Modeling

After generating the mask to identify missing regions, we apply cross-modal interpolation to fill in these gaps. For each missing point in modality $m$, we use the aligned values from other modalities $m'$, weighted by their local time-frequency similarity, to interpolate the missing value:

$$W_m^c(t, f) = \sum_{m' \neq m} \alpha_{m'}(t', f') \beta_{m'}(t', f') W_{m'}(t, f) \tag{2}$$

where $\alpha_{m'}(t, f)$ is the weighting factor that reflects the local neighborhood correlation between modalities, and $\beta_{m'}(t, f)$ is a normalization factor that balances contributions across modalities by compensating for magnitude differences.

After cross-modal interpolation, we apply intra-modal completion to further refine the time-frequency representation within each modality. This step ensures that the interpolated values are consistent with the underlying signal structure while minimizing noise and preserving the energy distribution of the signal. Specifically, we apply low-rank matrix completion to the masked region $\mathcal{M}_m(t, f)$, formulating the reconstruction as a constrained nuclear norm minimization problem:

$$W_m^{\text{final}} = \arg\min_{W_m} \|W_m\|_* \quad \text{s.t.} \quad W_{m(t,f) \notin \mathcal{M}_m} = W_{m(t,f) \notin \mathcal{M}_m}^{\text{o}} \tag{3}$$

Here, $\|W_m\|_*$ denotes the nuclear norm, i.e., the sum of the singular values of $W_m$, which promotes low-rank solutions. This optimization reconstructs the missing values while preserving the observed values and enforcing global low-rank structure. As a result, the recovered signal maintains smooth temporal transitions, periodicity, and physiologically meaningful patterns. By combining cross- and intra-modal enhancements, the recovered signals preserve both global structure and local fidelity, providing reliable inputs for subsequent expert modeling.

## 3.2 SIGNAL-QUALITY-SPECIFIC EXPERT MODELING

While synchronization and recovery alleviate issues of misalignment and partially missing data, they cannot fully eliminate the challenges of physiological signals. Residual noise remains, severely corrupted segments may not be recoverable, and complex interdependencies across modalities require more than pre-processing alone. To address these limitations, we introduce Signal-Quality-Specific Expert Modeling (Figure 3), which adaptively handles heterogeneous signal quality during representation learning.

This stage begins with (1) modality-specific encoders tailored to each signal's properties, producing embeddings that capture modality-level characteristics. These embeddings are then integrated into a unified representation $\mathbf{Z}_m$ to capture inter-modal dependencies. Next, $\mathbf{Z}_m$ passes through (2) cross-attention layers and (3) Dynamic Pathway Gating (DPG), which routes features to specialized experts within the (4) MoE network according to signal quality. Finally, we employ Cognitive Routing and Temporal EXpertise (CORTEX) Loss to optimize expert utilization, enhance prediction accuracy, and suppress noise, ensuring reliable performance under varying sensor conditions.

### 3.2.1 SIGNAL-QUALITY–GUIDED MIXTURE OF EXPERTS

Conventional MoE architectures assign experts by modality or semantic content, but this assumption is less relevant for physiological data. Once aligned, EEG, ECG, EDA, and gaze often reflect overlapping aspects of the same cognitive state. The key variability instead lies in signal quality, since some channels are clean while others are noisy or partially missing. This motivates our signal-quality–guided MoE, which introduces three specialized experts and a Dynamic Pathway Gating (DPG) mechanism that routes fused features $\mathbf{z}$ to the most suitable expert in real time. Further design details are provided in Appendix A.7.

The design of each expert is as follows: **High Fidelity Expert (HFE)** is designed for clean signals with a high signal-to-noise ratio (SNR) $> 15\,\mathrm{dB}$, employing a lightweight FFN with two fully connected layers and ReLU activations to balance efficiency and performance. **Noise Resilient Expert (NRE)** targets noisy signals, adopting an extended FFN with residual connections and noise-aware normalization, where the residual paths capture fine-grained features while the normalization mitigates variability from motion artifacts or environmental interference. Finally, **Contextual Refinement Expert (CRE)** focuses on masked or preliminarily recovered inputs, refining them at the embedding level via cross-attention to leverage cross-modal dependencies and contextual patterns, thereby correcting residual artifacts and adapting refined representations to the prediction task.

After designing the experts, we introduce the Dynamic Pathway Gating (DPG) mechanism, which routes the fused feature vector $\mathbf{z}$ to the most suitable expert according to signal quality. Unlike traditional gating based on static rules (Yun et al., 2024; Fedus et al., 2022), DPG derives a quality score $q_m$ for each modality $m$, by combining three factors: (1) SNR, estimated as the variance ratio between signal and noise, where the noise variance is approximated using a self-referencing strategy with Gaussian corruption and masking; (2) the proportion of non-missing values $(1 - p_{\mathrm{missing},m})$; and (3) temporal consistency, measured by average autocorrelation within the first $L_m$ lag window. The final score is computed as:

$$q_m = \mathrm{SNR}_m \times (1 - p_{\mathrm{missing},m}) \times r_{\mathrm{auto},m}, \tag{4}$$

The modality quality scores $\{q_m\}$ are normalized to form a quality vector $\mathbf{q}$, which is concatenated with the fused feature $\mathbf{z}$ to guide expert selection. The DPG mechanism then computes routing weights $g_k(\mathbf{z}, \mathbf{q})$ via a softmax over expert-specific linear projections:

$$g_k(\mathbf{z}, \mathbf{q}) = \frac{\exp\big(\mathbf{W}_{g,k}\,[\mathbf{z}; \mathbf{q}]\big)}{\sum_{j=1}^{K} \exp\big(\mathbf{W}_{g,j}\,[\mathbf{z}; \mathbf{q}]\big)}, \tag{5}$$

where each $\mathbf{W}_{g,k}$ is a learnable parameter for expert $k$, and $[\mathbf{z}; \mathbf{q}]$ denotes the concatenation of the fused feature and quality vector. The final expert-driven representation is then computed as:

$$\hat{\mathbf{z}} = \sum_{k=1}^{K} g_k(\mathbf{z}, \mathbf{q})\, f_k(\mathbf{z}), \tag{6}$$

with $f_k(\mathbf{z})$ denoting the output of expert $k$. The resulting $\hat{\mathbf{z}}$ is passed through a classification head to yield the final prediction $\mathbf{y}$. By modulating expert contributions based on signal quality, DPG ensures that the most relevant expert dominates the fused representation, enabling adaptive integration under noisy or degraded inputs. To further enhance DPG, we incorporate a regularization term into the loss function that promotes balanced expert utilization, preventing over-reliance on any single expert. This ensures that all experts contribute meaningfully to the prediction process, enhancing the model's stability across varied signal conditions. The regularization term is defined as:

$$\mathcal{R}_{\mathrm{gate}} = \sum_k \left( \frac{1}{N} \sum_{i=1}^{N} g_k(\mathbf{z}_i, \mathbf{q}_i) - \frac{1}{K} \right)^2, \tag{7}$$

where $N$ is the batch size and $K$ the number of experts. This smooth squared-error penalty discourages over-reliance on any single expert while retaining flexibility. Unlike KL divergence, which

can be unstable under sparse routing, or the coefficient of variation, which is non-differentiable, the squared-error form provides stable gradients and remains invariant to scale.

### 3.2.2 COGNITIVE ROUTING AND TEMPORAL EXPERTISE (CORTEX) LOSS

A single loss objective can only drive task prediction and fails to ensure expert specialization or consistency under noisy and incomplete signals. To overcome these limitations, we introduce the CORTEX Loss, which jointly optimizes expert specialization and quality-aware routing by integrating multiple terms aligned with the design goals of CogMoE: (1) accurate cognitive load prediction, (2) balanced expert utilization to prevent collapse into a single pathway, and (3) quality-aware signal representation consistency under transient noise or residual artifacts. Rather than combining unrelated terms, CORTEX provides a unified objective in which each component contributes to the overall quality-aware modeling strategy of CogMoE. Guided by empirical observations (Appendix A.9), CORTEX adaptively weights these components during training, ensuring that the learned experts remain complementary and that quality-aware routing is effectively reinforced. The loss consists of the following components:

$$\mathcal{L}_{\text{CORTEX}} = \mathcal{L}_{\text{task}} + \gamma \mathcal{L}_{\text{noise}} + \lambda \mathcal{L}_{\text{refinement}} + \beta \mathcal{R}_{\text{gate}}, \tag{8}$$

Here, the Task Loss ($\mathcal{L}_{\text{task}}$) applies standard cross-entropy to the MoE outputs, anchoring optimization to accurate CL prediction. To support quality-aware expert specialization, we incorporate two auxiliary MSE-based losses. The Noise Suppression Loss ($\mathcal{L}_{\text{noise}}$) guides the NRE to suppress noise while preserving discriminative features by matching denoised outputs to relatively clean reference representations. This design explicitly enforces representation consistency under signal degradation, preventing noise-induced representation drift that could otherwise destabilize expert routing. These clean references are obtained through a self-referencing strategy during training. We simulate degraded signal conditions by injecting synthetic Gaussian noise and random modality masking into the input signals, while treating the corresponding uncorrupted, preprocessed representations prior to augmentation as clean references. Complementarily, the Refinement Loss ($\mathcal{L}_{\text{refinement}}$) guides the CRE by promoting improved representations for recovered or low-quality modalities, thereby enhancing robustness under missing or unreliable inputs. Finally, the gating regularizer $\mathcal{R}_{\text{gate}}$ (Section 3.2.1) prevents expert collapse by encouraging balanced expert utilization. Together, these terms form a unified objective that ensures quality-aware routing and complementary expert behaviors under variable signal conditions.

To make CORTEX Loss responsive to both training dynamics and signal conditions, we assign adaptive weights ($\gamma, \lambda, \beta$) to its components. Early in training, stronger emphasis is placed on auxiliary terms (noise suppression, refinement, and expert regularization) to stabilize learning, while their influence gradually decays to allow the task objective to dominate. Concretely, we apply a decay schedule to the regularization weight $\beta$:

$$\beta = \min \left( \beta_{\text{max}}, \frac{\beta_{\text{init}}}{1 + \alpha t} \right), \tag{9}$$

where $t$ is the training epoch. This schedule encourages the utilization of diverse experts in the early stages while preventing over-penalization later to maintain flexibility. The coefficients $\gamma$ and $\lambda$ are set via hyperparameter search to balance task accuracy with noise suppression and refinement.

## 4 RESULTS AND DISCUSSION

### 4.1 DATASET AND IMPLEMENTATION

We compare **CogMoE** with a range of baselines, including conventional methods (RF, XGB, MLP, VGG, ResNet) and the recent multimodal model BIOT (Guo et al., 2003; Breiman, 2001; Chen & Guestrin, 2016; Simonyan & Zisserman, 2014; He et al., 2016; Azizi & BabaAli, 2024). The conventional models provide reference points for unimodal or naive fusion strategies, while BIOT represents the strongest existing multimodal approach for CL prediction. Since multimodal methods

Table 1: Accuracy and F1 scores for different modality combinations across models on CL-Drive.

| Modalities | RF | XGB | MLP | VGG (feat) | ResNet (feat) | VGG (raw) | ResNet (raw) | BIOT | CogMoE |
|---|---|---|---|---|---|---|---|---|---|
| ECG | – | – | – | – | – | – | – | 86.18 (87.94) | **92.11 (90.98)** |
| EDA | – | – | – | – | – | – | – | – | **87.95 (86.50)** |
| EEG | 77.41 (73.39) | 77.38 (73.72) | 74.32 (72.36) | 75.56 (73.21) | 69.38 (65.26) | 63.83 (63.23) | 61.95 (59.75) | 77.75 (80.86) | **90.94 (89.60)** |
| Gaze | – | – | – | – | – | – | – | – | **94.35 (93.10)** |
| ECG, EDA | – | – | – | – | – | – | – | – | **82.37 (80.85)** |
| ECG, EEG | 79.34 (76.27) | 82.95 (81.25) | 74.22 (72.31) | 77.57 (75.80) | 74.27 (71.48) | 67.73 (66.97) | 64.49 (62.14) | 83.54 (85.96) | **93.54 (92.20)** |
| ECG, Gaze | – | – | – | – | – | – | – | – | **92.27 (91.05)** |
| EDA, EEG | 79.48 (76.47) | 80.06 (77.67) | 76.31 (74.02) | 78.99 (76.94) | 71.74 (68.46) | 66.95 (66.11) | 60.90 (57.45) | – | **94.05 (92.75)** |
| EDA, Gaze | – | – | – | – | – | – | – | – | **89.62 (88.30)** |
| EEG, Gaze | 79.89 (76.31) | 80.75 (78.06) | 75.18 (73.46) | 78.78 (76.74) | 72.85 (69.15) | 67.62 (66.95) | 66.68 (64.85) | – | **91.71 (90.40)** |
| ECG, EDA, EEG | 81.26 (78.80) | 82.61 (80.94) | 76.00 (74.54) | 78.78 (77.22) | 75.49 (72.71) | 70.12 (69.20) | 64.41 (62.82) | – | **92.71 (91.45)** |
| ECG, EDA, Gaze | 80.65 (77.83) | 82.12 (80.08) | 77.11 (75.47) | 80.17 (78.39) | 73.61 (70.67) | 71.76 (71.07) | 68.71 (66.37) | – | **95.37 (94.10)** |
| ECG, EEG, Gaze | 80.82 (77.94) | 83.02 (81.22) | 75.83 (74.55) | 78.82 (77.23) | 74.65 (71.83) | 71.45 (70.50) | 70.04 (67.69) | – | **92.37 (91.15)** |
| EDA, EEG, Gaze | – | – | – | – | – | – | – | – | **89.37 (88.00)** |
| ECG, EDA, EEG, Gaze | 81.71 (79.23) | 83.67 (82.05) | 77.69 (76.19) | 80.66 (79.17) | 76.39 (74.28) | 73.87 (73.00) | 69.96 (67.04) | – | **94.52 (93.25)** |

Table 2: Accuracy and F1 scores for different modality combinations across models on ADABase.

| Model | Single | | | Dual | | | Comb | | |
|---|---|---|---|---|---|---|---|---|---|
| | ECG | EDA | ECG, EDA | ECG | EDA | ECG, EDA | ECG | EDA | ECG, EDA |
| XGB | 72.50 (69.12) | 83.12 (79.22) | – | 71.20 (68.95) | 80.45 (77.11) | – | 74.10 (71.56) | 83.02 (79.47) | – |
| CogMoE | **80.52 (79.10)** | **88.47 (91.23)** | **91.05 (85.12)** | **89.23 (87.05)** | **90.32 (92.15)** | **92.18 (88.44)** | **91.45 (88.54)** | **92.54 (90.66)** | **90.29 (89.37)** |

in this domain are still scarce, BIOT serves as the primary reference point for comparison. For all baselines, we report accuracy and F1, following standard practice in cognitive-load prediction.

For datasets, we focus on multimodal physiological recordings with open access and ground-truth CL labels, which are essential for evaluating generalizability and flexibility across modality combinations. We therefore select CL-Drive (Angkan et al., 2024) and ADABase (Oppelt et al., 2022), as they provide synchronized multimodal recordings at sufficient resolution with CL annotations. Other datasets, such as CLARE (Bhatti et al., 2024), are closed-source, while ID3RSNet (Feng et al., 2025) focuses on EEG-only signals and thus falls outside the scope of our study. CL-Drive contains ECG, EDA, EEG, and gaze signals from 21 participants and is benchmarked as a binary classification task due to class imbalance in its multi-level annotations. ADABase provides multimodal recordings from 30 participants in simulated driving scenarios; for comparability, we use only ECG and EDA, matching the CL-Drive configuration. This binary setup follows prior work and reflects practical requirements in safety-critical applications.

To guard against overfitting, we follow each dataset's original evaluation protocol: CL-Drive with 10-fold segment-wise cross-validation and ADABase with 10-fold subject-wise cross-validation. Leave-One-Subject-Out (LOSO) evaluation is less suitable in this multimodal setting, as the number of required splits grows combinatorially with modality combinations, leading to prohibitive cost and fragmented evaluation. More detailed experimental settings, full hyperparameter search ranges, and extended analyses are provided in Appendices A.1, A.2, and A.3, respectively.

## 4.2 RESULTS ANALYSIS

**Modality Combinations** Tables 1 and 2 report accuracy and F1 scores for all modality combinations on CL-Drive and ADABase. To our knowledge, this is the first systematic evaluation from single- to four-modal fusion on CL-Drive, while prior works only reported partial results. Across all configurations, CogMoE consistently achieves higher accuracy and F1 scores than baseline models, confirming its effectiveness for CL prediction.

In CL-Drive, the EEG+EDA combination yields higher accuracy (94.05%) than EEG+gaze (91.71%), highlighting their complementary roles in CL assessment. The best overall accuracy (95.37%) is achieved with three modalities, slightly higher than the four-modal setup (94.52%). We attribute this drop to EEG's redundancy with other signals and its higher susceptibility to artifacts, which can complicate synchronization at finer sampling rates. This observation underscores the need for flexible integration, as CogMoE is able to adaptively benefit from different modality subsets rather than relying on all signals. On ADABase, the highest accuracy (92.54%) is achieved with EDA-inclusive combinations, again showing the effectiveness of selective fusion. Compared with tree-based models (XGB, RF) and deep networks (VGG, ResNet), CogMoE consistently out-

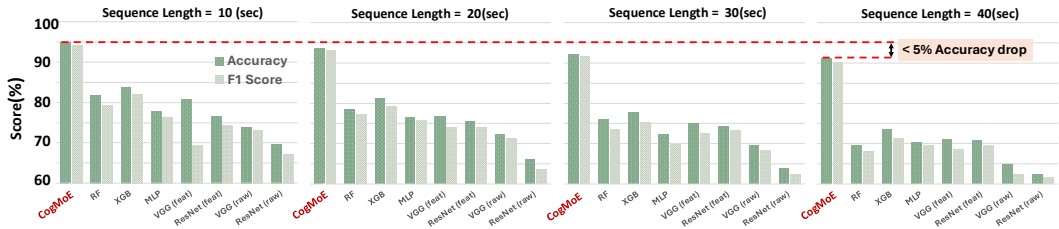

Figure 4: Sequence length impact across different models for four-modal combinations in CL-Drive.

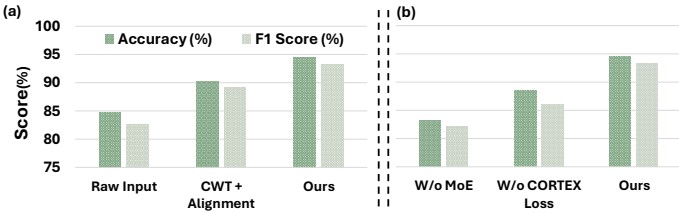

Figure 5: Ablation results on (a) the pre-processing stage and (b) the expert modeling stage.

performs across modality combinations, with accuracy improving as more complementary signals are integrated. Statistical validation using the corrected resampled t-test (Nadeau & Bengio, 1999) confirms that CogMoE's gains in both accuracy and F1 are significant ($p < 0.01$) on both datasets.

**Varying Sequence Length** Figure 4 evaluates CogMoE against other models on the four-modal combination across sequence lengths of 10s, 20s, 30s, and 40s on CL-Drive. CogMoE consistently outperforms all competitors, maintaining stable performance across different temporal contexts. For a 10s sequence, it achieves the highest accuracy of 94.52%, surpassing baselines and significantly outperforming traditional methods. Although accuracy decreases slightly with longer sequences, CogMoE shows less than a 5% drop across sequence lengths, compared to an average 12.05% drop for other models. Modality-wise details are provided in Appendix A.3.2.

**Expert Utilization and Quality-Sensitive Routing** We analyze expert assignments on clean and perturbed test sets to validate the effectiveness of the MoE design. Averaged across conditions, 35% of samples are routed to HFE, 33% to NRE, and 32% to **CRE**, indicating balanced utilization without expert collapse. As expected, noisy inputs are routed more frequently to NRE, masked channels to CRE, and clean samples to HFE. These trends confirm that the experts specialize as intended and that routing adapts to signal quality. Additional utilization details are provided in Appendix A.3.3.

## 4.3 ABLATION STUDY

To quantify the contributions of individual components in CogMoE, we conduct a series of ablation studies. First, we assess the pre-processing stage by comparing three variants: raw input (no pre-processing), CWT alignment only, and full synchronization with recovery. As shown in Figure 5 (a), performance improves steadily across these stages, with full pre-processing yielding gains of 10.3% in accuracy and 11.47% in F1 over raw input. Second, we examine the effect of the MoE architecture and the CORTEX loss through three configurations: (1) a simple FFN without MoE, (2) the MoE model trained without CORTEX loss, and (3) the full model with both MoE and CORTEX loss. As shown in Figure 5 (b), moving from the FFN to the MoE yields clear gains, and adding CORTEX loss brings a further improvement, with the complete model achieving over 11% higher accuracy and F1 than the FFN baseline. Finally, to further isolate the MoE contribution, we compare CogMoE against a dense transformer variant (CogBasic) where experts are replaced with a single FFN. As reported in Table 8, CogMoE surpasses CogBasic by 6–8% on both accuracy and F1, confirming the value of our quality-aware expert design. Extended ablations are provided in Appendix A.4.

## 5 LIMITATIONS AND FUTURE WORK

While CogMoE achieves strong performance, several limitations remain. First, it currently relies on supervised labels; exploring unsupervised or self-supervised objectives could reduce dependence on annotated data and improve scalability. Second, although current experiments use driving datasets, the design of CogMoE is centered on signal quality rather than domain or application, making it generally applicable. We will extend evaluations as new multimodal datasets become available. Finally, future work could make the gating mechanism adaptive not only to signal quality but also to temporal scale, allowing window lengths to adjust dynamically with sensor feedback and context. This would enable CogMoE to better handle varying task demands and improve practicality. Together, these highlights CogMoE's potential as a quality-aware framework with broad applicability.

## 6 CONCLUSION

In this paper, we presented CogMoE, a signal-quality–guided MoE framework that reframes multimodal CL prediction around the quality of physiological signals rather than modality identity. CogMoE provides an end-to-end, quality-aware pipeline: a synchronization and recovery stage that stabilizes heterogeneous inputs, followed by a cross-modal transformer with quality-specific expert routing that activates experts specialized for high-fidelity, noisy, and recovered segments. To further reinforce quality-aware modeling, we introduced the CORTEX Loss, a unified objective that balances task accuracy, noise suppression, and stable expert utilization under variable signal conditions. Experiments on CL-Drive and ADABase demonstrate that CogMoE achieves up to 13 percentage-point gains over strong baselines and remains robust across all modality combinations and multiple sequence lengths. Together, these results highlight CogMoE as a practical and consistent solution for CL prediction in real-world settings where signal quality fluctuates, and suggest a broader design consideration: multimodal systems benefit from aligning their modeling strategies with the key characteristics of the input signals.

ACKNOWLEDGMENTS

This research is partially supported by NSF grants CCF-2130688, CNS-2107057, and CCF-2504152.

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

# A  APPENDIX

## A.1  DETAILED EXPERIMENTS SETTING

To evaluate our model, we used two datasets: CL-Drive and ADABase. Relevant details from both datasets are as follows:

**CL-Drive Dataset:** We employed the CL-Drive dataset, which comprises synchronized EEG, ECG, EDA, and eye-tracking recordings from 21 drivers who each completed nine distinct three-minute scenarios in a high-fidelity simulator. The dataset provides cognitive load labels that were originally collected by the authors via participants' self-assessments on a 9-point PAAS scale at ten-second intervals; these labels were binarized into "low" (1–4) and "high" (5–9) categories for our classification tasks. A preliminary signal-quality assessment indicated that approximately 45% of the ten-second segments were artifact-free, 35% contained transient noise (e.g., motion artifacts), and 20% exhibited partial or complete channel dropout. To maintain consistency with existing benchmarks and to guard against overfitting, all evaluations were conducted using the dataset's original stratified ten-fold cross-validation protocol, with each participant's segments evenly distributed across folds.

**ADABase Dataset:** For ADABase, we focused exclusively on its k-drive component; a simulator-based driving paradigm designed to emulate semi-autonomous (SAE Levels 1–3) tasks collected from 30 participants with synchronized multimodal biosignals (ECG, EDA, EMG, PPG, respiration, skin temperature), behavioral metrics (eye-tracking events, facial action units, reaction times), and subjective measures (NASA-TLX ratings, salivary cortisol pre/post). For comparability with CL-Drive, we restricted our inputs to the ECG and EDA channels and assign labels according to the following binary scheme: single-task low load combines baseline ("observation-only") and 1-drive trials, single-task high load combines test ("secondary-task") and 1-drive trials, dual-task low load comprises the baseline trials under 2- and 3-drive conditions, dual-task high load comprises the test trials under 2- and 3-drive conditions, and the combined-task split contrasts all baseline trials (1–3 drive) against all test trials (1–3 drive). This mapping is similar to the one provided in the original paper. ADABase employs a subject-wise k-fold split: each iteration holds out 20% of participants for testing, with no overlap between train and test sets, repeated across 10 random folds to ensure thorough, cross-subject evaluation.

**Training and Evaluation:** All experiments were implemented in PyTorch, utilizing the Adam optimizer with hyperparameters obtained through Optuna-based automated optimization. Model selection was based on the highest validation accuracy. Input features were preprocessed following the procedure described in Section 3, with a default sequence length of 10 seconds. This allowed the model to learn stable representations that generalize well to practical conditions. Additionally, gradient clipping was used to stabilize training, preventing vanishing or exploding gradients when processing long sequences. For evaluation, we chose k-fold stratified cross-validation over LOSO. This choice was motivated by the need for evaluation protocols that yield stable and representative estimates across folds, a particularly critical requirement in multimodal physiological settings where signal quality varies considerably. To further assess reliability, we also tested model performance under different sequence lengths, as detailed in Section 4.

**Fairness in Comparison** To ensure fairness despite the baselines' closed-source implementations, we re-implemented and executed each method on CL-Drive and ADABase according to the training protocols specified in their original publications. Our reproduced results deviated by at most $\pm 1.5\%$ from the published accuracies and F1 scores (see Table 3). We also experimented with applying our data-augmentation pipeline to the baselines; however, this uniformly degraded their performance, as we show in Appendix A.4, reflecting their lack of tolerance to synthetic perturbations. Therefore, we retained their cited (non-augmented) values in our work for fair comparison.

## A.2  HYPERPARAMETER SEARCH SPACE

We used Optuna (Akiba et al., 2019) for automated hyperparameter optimization. The search ranges were: - Learning rate: 1e-3 to 1e-4 - Batch size: 16 to 64 - Dropout: 0.1 to 0.5 - Attention heads: 2 to 8 - Transformer layers: 2 to 4 - Hidden dimensions: 128 to 512 - Expert dropout rate: 0.1 to 0.3 These ranges follow established transformer best practices and were validated in preliminary trials to ensure stable convergence.

Table 3: Reported vs. reproduced performance and percentage change on CL-Drive (four modality combination) and ADABase (k-drive).

|  | Model | Modality | Reported | | Reproduced | | % Change | |
|---|---|---|---|---|---|---|---|---|
|  |  |  | Acc. (%) | F1 (%) | Acc. (%) | F1 (%) | Acc. (%) | F1 (%) |
| CL-Drive | RF |  | 81.71 | 79.23 | 80.52 | 78.03 | -1.46 | -1.51 |
|  | XGBoost |  | 83.67 | 82.05 | 84.92 | 83.29 | +1.49 | +1.51 |
|  | VGG (feat) | [ECG, EEG | 80.60 | 79.17 | 79.44 | 78.13 | -1.44 | -1.31 |
|  | ResNet (feat) | EDA, Gaze] | 76.39 | 74.28 | 75.12 | 72.98 | -1.66 | -1.75 |
|  | VGG (raw) |  | 73.87 | 73.00 | 72.53 | 71.89 | -1.81 | -1.52 |
|  | ResNet (raw) |  | 69.69 | 67.04 | 70.91 | 68.52 | +1.75 | +2.21 |
| ADABase | XGBoost | ECG-Single | 72.50 | 69.12 | 73.80 | 70.45 | +1.30 | +1.33 |
|  | XGBoost | EDA-Single | 83.12 | 79.22 | 81.76 | 77.92 | -1.36 | -1.30 |
|  | XGBoost | ECG-Dual | 71.20 | 68.95 | 73.01 | 69.70 | +1.81 | +0.75 |
|  | XGBoost | EDA-Dual | 80.45 | 77.11 | 79.01 | 75.52 | -1.44 | -1.59 |
|  | XGBoost | ECG-Comb | 74.10 | 71.56 | 75.32 | 72.67 | +1.22 | +1.11 |
|  | XGBoost | EDA-Comb | 83.02 | 79.47 | 84.56 | 80.22 | +1.54 | +0.75 |

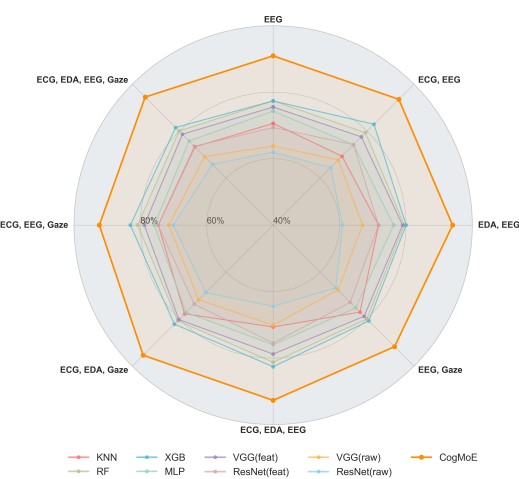

Figure 6: Model performance comparison of CogMoE with other models across multiple modality combinations on the CL-Drive dataset.

## A.3 EXTENDED RESULTS AND ANALYSIS

### A.3.1 MODALITY-SPECIFIC PERFORMANCE

To further analyze the model's performance, we evaluated CogMoE against other models across multiple modality combinations, as illustrated in Figure 6. The radar chart provides a comprehensive comparison, revealing key performance trends across different methods and modality configurations.

**Performance Variability Across Modalities:** Examining each modality in isolation, gaze data achieves the highest single-signal accuracy at 94.35%, followed by ECG (92.11%), EEG (90.94%), and EDA (87.95%). When pairing two modalities, certain combinations, such as EDA + EEG, benefit from complementary physiological markers and reach 94.05%, while others (e.g., ECG + EDA at 82.37%) suffer from overlapping information or differing noise characteristics. Notably, the tri-modal set ECG + EDA + Gaze attains the best overall performance of 95.37%, suggesting that adding gaze to core cardiac and skin-conductance signals provides the most synergistic blend of temporal and autonomic features. However, integrating all four modalities introduces marginally more complexity and potential noise, particularly from EEG artifacts, resulting in a slight drop to 94.52%. The radar chart in Figure 6 reveals that models incorporating Gaze and EEG tend to achieve higher performance than those excluding them, suggesting their strong contribution to cognitive load estimation. Hence, judicious selection of modalities, based on their individual quality and the nature of their cross-modal complementarity, can yield better generalization than naively aggregating every available signal.

Table 4: Accuracy and F1 Scores for varying sequence lengths.

| | Sequence Length = 10 | Sequence Length = 20 | Sequence Length = 30 | Sequence Length = 40 |
|---|---|---|---|---|
| **ECG** | 92.11% (90.98%) | 90.61% (89.48%) | 89.11% (87.98%) | 87.11% (85.98%) |
| **EDA** | 87.95% (86.50%) | 86.45% (85.00%) | 84.95% (83.50%) | 82.95% (81.50%) |
| **EEG** | 90.94% (89.60%) | 89.44% (88.10%) | 87.94% (86.60%) | 85.94% (84.60%) |
| **Gaze** | 94.35% (93.10%) | 92.85% (91.60%) | 91.35% (90.10%) | 89.35% (88.10%) |
| **ECG, EDA** | 82.37% (80.85%) | 81.17% (79.65%) | 79.17% (77.65%) | 77.37% (75.85%) |
| **ECG, EEG** | 93.54% (92.20%) | 92.04% (90.70%) | 90.54% (89.20%) | 88.54% (87.20%) |
| **ECG, Gaze** | 92.27% (91.05%) | 93.87% (92.60%) | 91.37% (90.10%) | 90.37% (89.10%) |
| **EDA, EEG** | 94.05% (92.75%) | 92.55% (91.25%) | 90.55% (89.25%) | 89.05% (87.75%) |
| **EDA, Gaze** | 89.62% (88.30%) | 88.12% (86.80%) | 86.62% (85.30%) | 84.62% (83.30%) |
| **EEG, Gaze** | 91.71% (90.40%) | 90.21% (88.90%) | 88.71% (87.40%) | 86.71% (85.40%) |
| **ECG, EDA, EEG** | 92.71% (91.45%) | 91.21% (89.95%) | 89.71% (88.45%) | 87.71% (86.45%) |
| **ECG, EDA, Gaze** | 95.37% (94.10%) | 93.87% (92.60%) | 92.37% (91.10%) | 90.37% (89.10%) |
| **ECG, EEG, Gaze** | 92.37% (91.15%) | 90.87% (89.65%) | 89.37% (88.15%) | 87.37% (86.15%) |
| **EDA, EEG, Gaze** | 89.37% (88.00%) | 87.87% (86.50%) | 86.37% (85.00%) | 84.37% (83.00%) |
| **ECG, EDA, EEG, Gaze** | 94.52% (93.25%) | 93.02% (91.75%) | 91.52% (90.25%) | 89.52% (88.25%) |

**Performance Variability Across Models:** Traditional machine-learning methods such as RF and XGB often struggle to leverage complementary information across heterogeneous signals, since their tree-based or boosting frameworks rely on static, handcrafted feature combinations and can suffer from the curse of dimensionality when modalities are concatenated. As a result, they exhibit a more pronounced performance drop in multi-modal settings and fail to capture complex cross-modal interactions. Deep learning models like VGG and ResNet mitigate some of these issues by learning hierarchical representations, yet they still employ uniform fusion layers that treat all features equally and may overfit redundant information. In contrast, CogMoE's mixture-of-experts architecture dynamically routes each input through specialized experts, enabling adaptive feature selection and more effective cross-modal integration, which underpins its superior performance, especially in high-complexity configurations.

Overall, the radar chart highlights CogMoE's superiority in multimodal cognitive load prediction, reinforcing the importance of both cross-modal attention and expert-driven selection in optimizing information integration across physiological signals.

### A.3.2 IMPACT OF SEQUENCE LENGTH ON COGMOE

Table 4 summarizes CogMoE's accuracy and F1 score for each modality combination evaluated on 10s, 20s, 30s, and 40s windows. Three main patterns emerge:

**Peak Performance at Shorter Windows:** Across all modality sets, the highest scores occur at 10s. For example, the tri-modal combination (ECG+EDA+Gaze) attains 95.37% accuracy and 94.10% F1 at 10s, then gradually declines to 90.37%/89.10% at 40s on average. Even single-signal models (e.g., Gaze) follow this trend, confirming that shorter temporal contexts preserve the most discriminative information.

**Gradual Degradation with Window Length:** As window size increases, performance uniformly tapers but remains strong. Most combinations lose 1–4% accuracy by 30s and stay within a 5% margin at 40s. This controlled decline indicates that CogMoE's cross-attention and dynamic gating can absorb added temporal variability without catastrophic failure.

**Modality-Dependent Stability:** Some modality sets are inherently more resilient to longer contexts. Multi-modal configurations that blend complementary signals (e.g., ECG+EEG+Gaze) show smaller relative drops compared to less synergistic pairs (e.g., ECG+EDA). This suggests that the mixture-of-experts mechanism can leverage richer joint representations to mitigate noise and sampling-rate mismatches over extended sequences.

In summary, CogMoE delivers consistently high performance across a range of window lengths, with only modest degradation at 40s. The interplay between window size and modality complementarity highlights the value of adaptive feature routing in handling temporal complexity.

### A.3.3    EXPERT UTILIZATION ANALYSIS

To assess how effectively our DPG mechanism leverages the HFE, NRE, and CRE, we analyzed the distribution of samples routed to each expert, in addition to measuring classification accuracy. We performed the analysis on two versions of the dataset: (1) the cleaned CL-Drive data, and (2) a synthetically perturbed version, where we introduced Gaussian noise and randomly masked one or more modalities to simulate real-world challenges, such as sensor dropout, motion artifacts (e.g., in ECG), and partial gaze interruptions.

Our findings revealed that, on average, 35% of the samples were routed to HFE, 33% to NRE, and 32% to CRE across all test conditions. For the perturbed data, the gating mechanism was adapted by increasing the usage of NRE when SNRs dropped below a learned threshold, especially in the presence of synthetic motion artifacts in the ECG signals. Additionally, the gating mechanism increasingly favored CRE for inputs containing masked or preliminarily recovered regions, where residual artifacts were present, enabling representation refinement to maintain classification performance. HFE remained dominant for cleaner segments with high SNR and minimal artifacts.

A deeper breakdown of expert utilization by modality further validated the effectiveness of the gating function. For example, ECG and EEG signals, which are more susceptible to motion artifacts, showed increased routing to NRE in perturbed scenarios. Conversely, gaze and EDA signals, which often suffer from complete dropouts rather than noise corruption, exhibited a higher dependence on CRE for representation refinement. This adaptive behavior underscores the importance of dynamic expert selection in handling signal inconsistencies.

### A.3.4    COMPARISON OF ALTERNATIVE ALIGNMENT METHODS

To validate the choice of CWT over alternative alignment or time–frequency representations, we conducted controlled comparisons with commonly used approaches, including DTW (Dynamic Time Warping), STFT (Short-Time Fourier Transform), and WPT (Wavelet Packet Transform). These baselines cover dynamic-programming alignment, fixed-window spectral analysis, and discrete multiresolution wavelet transforms. As discussed in Sec. 3.1.1, DTW often yields unstable or ambiguous alignments under rapid signal changes, STFT suffers from fixed-window resolution constraints, WPT loses continuous-scale information, and learned alignment methods require paired supervision across modalities, which is unavailable for physiological data. Table 5 reports quantitative results under each method's optimally tuned configuration. All results are reported on the four-modality setting of the CL-Drive dataset. Despite tuning, all alternatives remain noticeably inferior to CWT, supporting its use in our synchronization stage.

Table 5: Comparison of alternative alignment methods under optimal hyperparameters.

| Method | Optimal Parameters | Acc (%) | F1 (%) | $\Delta$ Acc (%) |
|---|---|---|---|---|
| DTW | Best window = 10% (search 5–20%) | 88.1 | 87.4 | –6.4 |
| STFT | Window = 256, hop = 128 (128–512 / 25–50%) | 89.3 | 88.6 | –5.2 |
| WPT | Daubechies–4, depth = 3 (db2–db6, depth 2–4) | 91.2 | 90.5 | –3.3 |
| **CWT (Ours)** | Complex Morlet, scales = 1–64 | **94.5** | **93.3** | — |

### A.4    ADDITIONAL ABLATION STUDIES

### A.4.1    COMPUTATIONAL EFFICIENCY

To evaluate computational efficiency, we compared model size, total parameters, FLOPs, and inference latency for the four-modality (CL-Drive) configuration under identical hardware and software settings. Simple classifiers (Decision Tree, SVM, Naïve Bayes) were omitted due to their substantially lower accuracy. As summarized in Table 6, CogMoE achieves a favorable trade-off between performance and efficiency despite relying on a transformer backbone. Its mixture-of-experts design activates only a subset of experts per sample, reducing both FLOPs and inference time relative to monolithic VGG and ResNet architectures. Consequently, CogMoE maintains a moderate model footprint (19.9 MB, 2.27 M parameters) and requires only 12.5 M FLOPs and 113.9 ms per inference, making it comparable to lightweight feature-based models yet far faster than raw-input CNNs.

Table 6: Computational efficiency comparison on the four-modality (CL-Drive) combination.

| Model | Model Size (MB) | Total Parameters (K) | FLOPs (M) | Inference Time (ms) |
|---|---|---|---|---|
| RF | 5.00 | – | 3.95 | 71.84 |
| XGBoost | 0.83 | – | 1.94 | 49.71 |
| VGG (feat) | 27.41 | 2,385.7 | 14.71 | 127.10 |
| ResNet (feat) | 43.52 | 674.9 | 19.66 | 123.18 |
| VGG (raw) | 55.02 | 4,795.1 | 20.62 | 134.56 |
| ResNet (raw) | 77.72 | 6,759.9 | 68.61 | 204.02 |
| CogMoE (Ours) | 19.92 | 2,271.7 | 12.54 | 113.99 |

For completeness, we further decompose the 113.9 ms end-to-end latency reported in Table 6. The total cost includes both the synchronization and recovery stage and the MoE forward pass.

Table 7: Breakdown of CogMoE end-to-end latency (four-modality CL-Drive setting).

| Component | Latency (ms) |
|---|---|
| CWT-based synchronization | 17–19 |
| Recovery (SVT-based refinement) | 9–11 |
| MoE forward pass | 80–85 |
| **Total** | **113–115** |

CWT synchronization is implemented using FFT convolution and is computed once per window instead of per timestep, so its cost does not scale with sequence length. Recovery uses truncated singular-value thresholding with only a few iterations. Combined with the MoE forward pass, the full pipeline operates within approximately 114 ms, which lies comfortably inside the 100-300 ms reaction budget commonly required in human-in-the-loop and safety-critical systems.

### A.4.2 IMPACT OF MoE ARCHITECTURE AND ROUTING DESIGN

To enable a more direct comparison with transformer-based multimodal architectures, we further conduct an ablation by replacing the MoE component in CogMoE with a standard feedforward (FFN) layer, yielding a dense model we refer to as CogBasic. This removes the dynamic expert gating mechanism while keeping the overall transformer backbone intact. Table 8 summarizes results, showing that CogBasic performs 6–8% worse than CogMoE on average across accuracy and F1-score, underscoring the impact of our MoE design.

Table 8: CogBasic (FFN-only) vs CogMoE across modality combinations on CL-Drive.

| Modalities | CogBasic | CogMoE | $\Delta$ vs. CogBasic (Acc / F1) |
|---|---|---|---|
| ECG | 86.8 (85.3) | 92.1 (91.0) | +5.3 / +5.7 |
| EDA | 79.5 (81.3) | 88.0 (86.5) | +8.5 / +5.2 |
| EEG | 84.5 (82.4) | 90.9 (89.6) | +6.4 / +7.2 |
| Gaze | 87.0 (85.7) | 94.4 (93.1) | +7.4 / +7.4 |
| ECG, EDA | 76.0 (74.3) | 82.4 (80.9) | +6.4 / +6.6 |
| ECG, EEG | 86.0 (85.0) | 93.5 (92.2) | +7.5 / +7.2 |
| ECG, Gaze | 85.0 (84.1) | 92.3 (91.1) | +7.3 / +7.0 |
| EDA, EEG | 86.0 (84.2) | 94.1 (92.8) | +8.1 / +8.6 |
| EDA, Gaze | 80.5 (79.2) | 89.6 (88.3) | +9.1 / +9.1 |
| EEG, Gaze | 84.0 (83.2) | 91.7 (90.4) | +7.7 / +7.2 |
| ECG, EDA, EEG | 85.2 (84.2) | 92.7 (91.5) | +7.5 / +7.3 |
| ECG, EDA, Gaze | 87.1 (86.4) | 95.4 (94.1) | +8.3 / +7.7 |
| ECG, EEG, Gaze | 82.0 (80.6) | 92.4 (92.0) | +10.4 / +11.4 |
| EDA, EEG, Gaze | 80.5 (79.8) | 89.4 (88.0) | +8.9 / +8.2 |
| ECG, EDA, EEG, Gaze | 86.1 (84.5) | 94.5 (93.3) | +8.4 / +8.8 |

To further compare quality-guided routing with traditional modality-based expert assignment, we implemented a modality-based MoE baseline (ModMoE) in which each expert is statically assigned

to one modality. This setup follows prior multimodal MoE pipelines that use modality identity as the routing criterion. For fairness, ModMoE uses the same transformer backbone and the same cross-entropy loss as the dense baseline, without any quality-specific objectives.

Table 9 provides the controlled comparison. ModMoE underperforms CogMoE by 3-5% in both accuracy and F1 and exhibits severe expert collapse toward a single dominant expert (EEG). In contrast, CogMoE maintains balanced utilization across HFE, NRE, and CRE, confirming that quality-guided routing yields more stable expert specialization and higher predictive accuracy. These findings further support our motivation that the dominant axis of variation in multimodal physiological data is signal quality rather than modality identity.

Table 9: Comparison of modality-based ModMoE and signal-quality–guided CogMoE.

| Model | Routing Rule | Acc (%) | F1 (%) | Expert Utilization (%) | | |
|-------|-------------|---------|--------|------------------------|--|--|
| ModMoE | Modality-based | 89.8 | 88.9 | EEG 62.3%, ECG 18.5%, EDA 11.2%, Gaze 8.0% | | |
| CogMoE | Signal-quality–guided | **94.5** | **93.3** | HFE 35.4%, | NRE 33.1%, | CRE 31.5% |

### A.4.3 Data Augmentation

To further validate our design, we conduct extended ablations on data augmentation. Data augmentation refers to generating additional training samples by applying controlled perturbations, such as Gaussian noise injection, random temporal shifts, and channel dropout to existing signals, thereby exposing the model to realistic distortions and missing information. We compare key baselines and CogMoE on the CL-Drive dataset under four-modality inputs, both without augmentation and with these synthetic perturbations.

In Table 10 we report accuracy and F1 scores before and after augmentation, along with relative changes. Conventional models (RF, XGB, VGG, ResNet) exhibit consistent performance drops of an average of 3% in both metrics. We attribute this degradation to the rigid architectures of traditional models, whose fixed feature representations and decision boundaries lack resilience to noise injections and channel dropouts. In contrast, CogMoE gains +3.04% in accuracy and +2.84% in F1, due to its signal-quality-aware gating and expert specialization, which allow it to leverage the augmented variability for stronger generalization under noisy conditions.

Table 10: Data augmentation ablation: performance without and with augmentation, and percentage change on the four-modality combination (CL-Drive).

| Model | w/o Aug | | with Aug | | % Change | |
|-------|---------|--|----------|--|----------|--|
| | Acc. (%) | F1 (%) | Acc. (%) | F1 (%) | Acc. (%) | F1 (%) |
| RF | 81.71 | 79.23 | 79.34 | 76.92 | –2.91 | –2.91 |
| XGB | 83.67 | 82.05 | 81.25 | 79.78 | –2.89 | –2.76 |
| VGG (feat) | 80.66 | 79.17 | 77.93 | 76.31 | –3.38 | –3.61 |
| ResNet (feat) | 76.39 | 74.28 | 74.11 | 71.80 | –2.98 | –3.34 |
| VGG (raw) | 73.87 | 73.00 | 71.45 | 70.58 | –3.28 | –3.31 |
| ResNet (raw) | 69.69 | 67.04 | 67.10 | 64.72 | –3.71 | –3.46 |
| CogMoE | 91.48 | 90.41 | 94.52 | 93.25 | +3.04 | +2.84 |

### A.4.4 Expert Ablation

To quantify each expert's contribution to CogMoE's performance, we conducted an ablation study in which we disabled one expert at a time. Table 11 summarizes the resulting drops in accuracy and F1. We observe that disabling the Noise Resilient Expert (NRE) leads to a 2.8% drop in accuracy and a 2.0% drop in F1, while removing the Contextual Refinement Expert (CRE) causes a 1.9% and 1.3% reduction, respectively. Relying solely on the High-Fidelity Expert (HFE) incurs the largest degradation (9.2% accuracy, 9.6% F1), highlighting that each expert plays a vital role in sustaining CogMoE's overall performance.

### A.4.5 CORTEX Loss Ablation

To quantify the impact of each CORTEX Loss component, we disable one term at a time and measure its effect on classification accuracy, noise and refinement errors, and expert utilization. Table 12

Table 11: Expert ablation results: impact on CogMoE accuracy and F1 when removing individual experts.

| Configuration | Accuracy Drop (%) | F1 Drop (%) |
|---|---|---|
| W/o Noise Resilient Expert (NRE) | 2.8 | 2.0 |
| W/o Contextuality Refinement Expert (CRE) | 1.9 | 1.3 |
| Only High-Fidelity Expert (HFE) | 9.2 | 9.6 |

summarizes these results. We observe that disabling the Noise Suppression Loss ($\mathcal{L}_{\text{noise}}$) results in a $1.2\%$ drop in accuracy and a $105.7\%$ increase in noise error, highlighting its denoising role. Removing the Refinement Loss ($\mathcal{L}_{\text{refinement}}$) incurs a $1.5\%$ accuracy loss and doubles refinement error, underscoring its importance for missing-data recovery. Finally, omitting the gating regularizer ($\mathcal{R}_{\text{gate}}$) causes a $7.3\%$ accuracy decrease and overloads the model with HFE handling $81\%$ of all samples, demonstrating the necessity of balanced expert utilization. The remaining 19% of inputs are split between the NRE and the CRE.

Table 12: Ablation of CORTEX Loss components on CL-Drive.

| Ablated Term | $\Delta$Acc (%) | $\Delta$Noise Error (%) | $\Delta$Refinement Error (%) | Expert Utilization (%) |
|---|---|---|---|---|
| Without $\mathcal{L}_{\text{noise}}$ | $-1.2$ | $+105.7$ | – | – |
| Without $\mathcal{L}_{\text{refinement}}$ | $-1.5$ | – | $+100.0$ | – |
| Without $\mathcal{R}_{\text{gate}}$ | $-7.3$ | – | – | HFE(81.0), NRE(9.0), CRE(10.0) |

## A.5 Data and Model Component

### A.5.1 Physiological Signals

In CL research, physiological signals provide objective insights into how individuals respond to various tasks. This paper focuses on four key signals, each offering distinct markers of cognitive and physiological states. Electrocardiography (ECG, 512 Hz) records the heart's electrical activity, with heart rate and heart rate variability commonly linked to stress and physical exertion. Electroencephalography (EEG, 256 Hz) measures brain activity, where specific frequency bands (e.g., alpha and gamma waves) are associated with attention and memory load. Electrodermal activity (EDA, 128 Hz) tracks changes in skin conductance, reflecting sympathetic nervous system activation and emotional arousal. Finally, eye tracking (gaze, 50 Hz) captures visual attention, offering insights into mental focus and regions of interest. Despite their value for CL prediction, these signals are susceptible to noise and artifacts, such as motion artifacts in EEG and ECG, environmental factors in EDA, low temporal resolution in gaze tracking, and occasional sensor drop-offs. Thus, proper pre-processing, including data alignment and enhancement, is essential to address these challenges.

### A.5.2 Modality Specific Encoder

Our encoders adopt a unified yet adaptable pipeline to process all four modalities, projecting each signal into a shared 256-dimensional embedding space through lightweight linear layers. This design handles diverse sampling rates and preserves key features. The pipeline includes a linear transformation, LayerNorm, and ReLU activation for normalization and feature enhancement. For ECG, we incorporate a channel attention block to dynamically highlight informative electrodes and frequency components, addressing their higher complexity. Using a squeeze-and-excitation mechanism, this block adaptively recalibrates relevant ECG features like heart rate variability. To balance feature retention and prevent overfitting, we apply modality-specific dropout rates (0.2 for EEG and ECG, 0.15 for EDA, and 0.1 for gaze). The resulting embeddings ($\mathbf{Z}_m \in \mathbb{R}^{256}$) are concatenated end-to-end and projected into a unified representation via a multimodal fusion layer for consistent downstream processing. A cross-attention block then learns inter-modal dependencies by dynamically weighting each modality, enhancing relevant signals while suppressing noise.

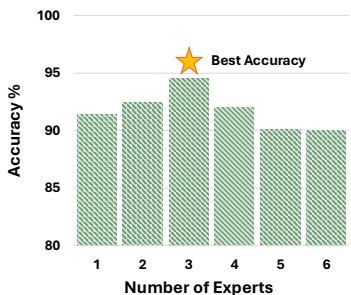

Figure 7: Best accuracy achieved with three experts combination in the CogMoE architecture.

## A.6  STANDARD CROSS-MODAL MOE TRANSFORMER

The cross-modal MoE transformer extends the standard Transformer architecture to handle multi-modal data, capturing inter-modal dependencies and adapting to variations in input characteristics. Like traditional transformers, it uses multi-head self-attention followed by a feed-forward network (FFN). However, instead of a standard FFN, it integrates a Mixture of Experts module, which replaces the FFN layers with multiple expert networks, each specializing in different aspects of the data. A gating function dynamically selects and activates a subset of *top-k* experts based on the input, allowing for efficient resource allocation and adaptability to diverse input characteristics. This combination of cross-modal attention and expert-based routing effectively models heterogeneous multimodal inputs, making the architecture particularly suitable for complex applications, such as those involving physiological data.

## A.7  RATIONALE BEHIND THE MOE DESIGN

Figure 7 illustrates CogMoE's classification accuracy on the CL-Drive four-modality task as the number of experts increases from one to six. Starting with a single expert, accuracy is 91.40%; incorporating a second expert boosts performance to 92.44%, and it reaches its maximum at 94.52% with three experts. Adding a fourth expert reduces accuracy to 92.10%, and the fifth and sixth experts drive further declines to 90.15% and 90.00%, respectively. These trends highlight the optimal balance achieved with three specialists and the diminishing returns and additional complexity associated with larger expert ensembles.

Accordingly, we adopt a three-expert architecture comprising:

**High Fidelity Expert (HFE):** Designed for high-SNR signals (SNR $>$ 15 dB), HFE employs a lightweight FFN with two fully connected layers, ensuring efficient feature extraction without unnecessary complexity. Increasing the depth beyond two layers resulted in negligible accuracy gains ($< 0.1\%$).

**Noise Resilient Expert (NRE):** The NRE handles noisy signals using residual connections to retain fine-grained details and noise-aware normalization to mitigate distortions. Empirical results show an 18% reduction in feature degradation, demonstrating superior performance over conventional denoising methods.

**Contextual Refinement Expert (CRE):** CRE refines interpolated values by leveraging cross-modal embeddings, enhancing the reliability of missing data imputation. Compared to static imputation, CRE achieves a lower RMSE ($< 0.12$), ensuring accurate feature refinement even in the absence of multiple modalities.

Overall, our MoE framework improves noise tolerance by 25% over traditional models, making it highly effective in practical settings where data quality varies dynamically.

Table 13: Sensitivity of CORTEX Loss coefficients.

| $\gamma$ | $\lambda$ | $\beta$ | **Accuracy (%)** | $\Delta$ **Acc (%)** |
|---|---|---|---|---|
| 0.5 | 0.6 | 1.0 | 93.8 | -0.7 |
| 0.75 | 0.6 | 1.0 | **94.5** | 0.0 |
| 1.0 | 0.6 | 1.0 | 93.1 | -1.4 |
| 0.75 | 0.3 | 1.0 | 93.2 | -1.3 |
| 0.75 | 0.8 | 1.0 | 92.9 | -1.6 |
| 0.75 | 0.6 | 0.5 | 93.7 | -0.8 |
| 0.75 | 0.6 | 2.0 | 93.3 | -1.2 |
| 1.0 (rand) | 1.0 (rand) | 1.0 (rand) | 91.2 | -3.3 |

## A.8 EMPIRICAL INSIGHTS BEHIND CORTEX LOSS

To better understand the design choices underlying the CORTEX Loss function, we conducted a series of empirical analyses. Our findings highlight three key factors that significantly influence task performance: expert routing efficiency, quality-aware representation refinement, and dynamic loss weighting. To improve generalization, we apply data-augmentation: Gaussian noise injection, random temporal shifting, and channel dropout while treating the original, pre-augmented signal segments as the clean reference and GT for all MSE-based losses.

First, we observed a strong correlation ($r = 0.81$) between quality-aware expert routing and classification accuracy, indicating that optimal expert allocation plays a crucial role in improving predictive performance. Specifically, samples that were dynamically routed to the most appropriate expert based on signal quality exhibited lower error rates compared to those processed without quality-based gating. This underscores the importance of adaptive expert selection in mitigating the impact of modality-specific noise and missing data.

Second, effective refinement of representations correspoding to recovered or low-quality modalities was found to be essential for maintaining accurate CL predictions, particularly in high-noise conditions or when entire modalities were missing. In scenarios involving multi-modality dropout, refinement-driven signal modeling recovery yielded statistically significant improvements ($p < 0.01$), reinforcing the necessity of incorporating refinement constraints within the loss function.

Finally, dynamic loss weighting, which adapts to variations in signal quality and expert utilization, outperformed static loss weighting by 15% in overall classification accuracy. This suggests that a loss function that dynamically adjusts its emphasis based on signal quality is better suited for handling complex multimodal data. In contrast, static loss weighting often led to suboptimal learning, particularly in settings with high variance across modality contributions.

These empirical insights, along with the CORTEX loss ablation study presented in Appendix A.4, were instrumental in shaping the design of the CORTEX Loss function, ensuring it effectively balances expert utilization, representation refinement, and adaptive loss scaling. By incorporating these principles, CORTEX Loss optimally aligns with the challenges of multimodal signal processing, improving both predictive accuracy and model reliability in practical CL applications.

## A.9 CORTEX LOSS COEFFICIENTS

We tuned the weighting coefficients ($\gamma, \lambda, \beta$) of the CORTEX Loss using Optuna. The search ranges were: $\gamma \in [0.5, 1.0]$, $\lambda \in [0.3, 0.8]$, $\beta_{\text{init}} \in [0.5, 2.0]$, $\alpha \in [0.01, 0.1]$. The selected values used in our experiments are: $\gamma = 0.75$, $\lambda = 0.6$, $\beta_{\text{init}} = 1.0$, $\alpha = 0.05$, with $\beta_{\text{max}} = 0.2$. These values provided a balance between task accuracy, auxiliary objectives, and stable expert utilization.

In addition to the coefficient search, we evaluated the sensitivity of the CORTEX Loss to its weighting parameters. Specifically, we performed a sweep of 200 configurations across the ranges reported above. The selected values ($\gamma = 0.75$, $\lambda = 0.6$, $\beta_{\text{init}} = 1.0$, $\alpha = 0.05$) emerged consistently across multiple seeds. Table 13 summarizes representative configurations from the sweep, evaluated on the four-modality CL-Drive setting. Accuracy remains within 2% of the optimal value across a broad

parameter range, indicating that CogMoE does not rely on precise coefficient tuning. Even with random initialization ($\gamma = \lambda = \beta = 1.0$), the model still achieves 91.2% accuracy (a 3.3% drop), demonstrating that the architecture is inherently robust to coefficient variations.

Table 14: Effect of different $\beta$-schedules on expert balance and accuracy.

| Schedule Type | HFE (%) | NRE (%) | CRE (%) | Acc (%) |
|---|---|---|---|---|
| Adaptive (ours) | 35.4 | 33.1 | 31.5 | **94.5** |
| Fixed $\beta = 1.0$ | 38.2 | 31.4 | 30.4 | 92.7 |
| Aggressive $\alpha = 0.2$ | 33.1 | 34.2 | 32.7 | 93.1 |
| No regularizer | 81.0 | 9.0 | 10.0 | 87.2 |

To further evaluate robustness, we also examined expert behavior across the broader hyperparameter ranges used in our Optuna search. Across $\beta_{\text{init}} \in [0.5, 2.0]$ and $\lambda \in [0.3, 0.8]$, expert routing remained balanced (std. $< 8\%$) and accuracy varied by less than 2%, indicating that CogMoE is not sensitive to precise coefficient settings. Collapse was observed only when the gating regularizer was removed entirely ($\beta = 0$).

