# OpenReview forum: "CogMoE: Signal-Quality–Guided Multimodal MoE for Cognitive Load Prediction"
_ICLR.cc/2026/Conference — ICLR 2026 Poster_

### Official Review · Reviewer_gQsN · 2025-10-25

**Soundness:** 2
**Presentation:** 2
**Contribution:** 3
**Rating:** 4
**Confidence:** 4

**Summary:**

The manuscript presents a potentially valuable idea emphasizing signal quality during multimodal fusion but currently falls short in exposition, motivation, and experimental rigor, evaluation.

**Strengths:**

CoGMoE / CogMoE

**Weaknesses:**

- Manuscript readability and presentation require improvement; the method is not explained in sufficient technical detail to assess reproducibility or novelty.

- Terminology is inconsistent (e.g., CoGMoE vs CogMoE); a single canonical name and notation should be used throughout.

- The problem statement and its significance are not clearly articulated; it is difficult to judge the scope and impact of the contribution.

- Related work discussion is insufficient: prior techniques and their specific limitations are not described in depth, leaving motivation for a new method unclear.

- Evaluation is limited in scope, both in metrics and experiments, preventing strong conclusions about effectiveness and generalizability.

**Questions:**

- Justify the claim: “However, the fundamental bottleneck is not the lack of sensors or models, but the variable quality of physiological signals.” What empirical evidence or literature supports this assertion?

- Provide motivation: Why was the decision made to “shift the focus of multimodal fusion to signal quality, the true bottleneck in practical CL prediction”? How does this design choice address limitations of prior fusion methods?

---

> ### Author Response · Authors · 2025-11-20
>
> We thank the reviewer for the feedback. We address each point below with direct evidence and citations to the submitted manuscript.
>
> ## Questions
>
> ### 1. Evidence for the bottleneck claim
> Real-world physiological signals **degrade unpredictably** due to motion, electrode drift, and sensor dropout [1,2]. Moreover, unlike traditional multimodal setups where modalities provide complementary information, EEG, ECG, EDA, and gaze in CL prediction are **redundant views of the same cognitive process** [3], making modality-based routing ineffective when modalities differ in quality.
>
> Our controlled comparison further confirms this: a modality-based MoE collapses to whichever modality happens to be cleanest and underperforms CogMoE by 3–5%, indicating that signal quality is the dominant limiting factor in multimodal CL prediction.
>
> ### 2. Motivation for quality-driven fusion
> Prior multimodal fusion methods route or weight inputs based on modality identity, implicitly assuming that each modality is clean, stable, and always informative. When physiological signal quality diverges across modalities, due to noise, misalignment, or missing segments, these assumptions break down, and the fusion mechanism becomes unreliable.
>
> CogMoE addresses this limitation by **routing experts based on estimated signal quality rather than modality identity**, enabling the model to adaptively down-weight degraded channels and leverage cleaner ones. This design directly addresses the shortcomings of prior fusion techniques and is empirically validated in our experiments, where quality-aware gating yields consistent gains over modality-based fusion.
>
> ## Weaknesses
>
> ### 1. Presentation and Technical Details
> Sec. 3.1 and 3.2 present complete mathematical definitions of synchronization, recovery, gating, and expert specialization. Full hyperparameter ranges and implementation details appear in Appendices A.1–A.4.  **Code will be released upon acceptance.**
>
> ### 2. Naming Consistency
> We verified the full source and found no occurrences of “CoGMoE.” Section titles are automatically capitalized by the template (e.g., “COGMOE” in headers), which may have caused visual confusion. **All in-text occurrences consistently use “CogMoE”.**
>
> ### 3. Problem Statement, Significance, and Contribution
> The problem formulation, significance, and contributions are already described in **Sections 1 and 2** of the manuscript. These sections introduce the core challenge of heterogeneous, time-varying physiological signal quality, explain its practical significance in safety-critical CL prediction, and motivate why prior multimodal and MoE-based methods fail under such conditions. **Our main contribution is shifting multimodal fusion from modality-based to signal-quality–based modeling, which directly addresses this problem setup.**
>
> ### 4. Related Work Coverage and Motivation
> The motivation and limitations of prior approaches are already analyzed in last sentences of Sections 2.1 and 2.2. Section 2.1 concludes by explaining why existing CL pipelines break under noise, misalignment, and missing segments, which motivates a quality-aware formulation. Section 2.2 concludes by noting that prior MoE methods route experts by modality identity and implicitly assume reliable inputs, an assumption that does not hold for physiological signals. These sections jointly provide the motivation and limitation analysis that the reviewer requests.
>
> ### 5. Evaluation Scope, Coverage, and Metrics
> The submission already provides a **broad and detailed evaluation** of CogMoE across effectiveness, robustness, and model behavior. Accuracy and F1 are the **standard metrics** used in prior cognitive-load prediction work, and our comparisons follow this established practice.
>
> We evaluate across all modality combinations, multiple sequence lengths, pipeline efficiency, extensive ablations, expert-routing analysis, and robustness to corruption (Tables 1–9, Fig. 5). To our knowledge, this is the **most complete multimodal evaluation** reported on CL-Drive and ADABase to date; prior works typically examine only a small subset of modality configurations.
>
> Regarding generalizability, only two public datasets provide synchronized EEG, ECG, EDA, and gaze suitable for multimodal CL prediction. CogMoE is therefore evaluated on all existing datasets that match the problem setting. Section 5 further discusses potential extensions to additional physiological domains as suitable datasets emerge.
>
> ## References:
> [1] Anwer, S. et. al (2024). Evaluation of data-processing and artifact-removal methods for wearable physiological signals.
>
> [2] Giangrande, A., et. al (2024). Motion artifacts in dynamic EEG recordings: Experimental observations, electrical modelling, and design considerations.
>
> [3] Martínez Vásquez, D. A. et. al (2023). Mutual information between EDA and EEG in multiple cognitive tasks and sleep deprivation conditions.

---

### Official Review · Reviewer_k4Qj · 2025-10-29

**Soundness:** 4
**Presentation:** 4
**Contribution:** 3
**Rating:** 8
**Confidence:** 4

**Summary:**

This paper addresses the critical challenge of real-world cognitive load (CL) prediction, where physiological signal quality is often poor and variable. The authors propose **CogMoE**, a novel Mixture-of-Experts (MoE) framework that, unlike traditional models, dynamically routes data based on signal quality rather than modality. The two-stage framework first performs quality-aware synchronization and recovery of multimodal signals (EEG, ECG, etc.). Then, a cross-modal transformer routes representations to one of three experts specialized for different quality regimes: a High Fidelity Expert (HFE) for clean signals, a Noise Resilient Expert (NRE) for noisy ones, and a Contextual Refinement Expert (CRE) for recovered data . The model is trained with a novel **CORTEX Loss**, which jointly optimizes for task accuracy, balanced expert utilization, and quality-aware objectives. Experiments on the CL-Drive and ADABase datasets show state-of-the-art performance, with accuracy gains of up to 13% and 9.5%, respectively.

**Strengths:**

1. Novelty of Core Concept: The paper's central idea of a "signal-quality-guided" MoE is highly novel and directly targets the primary bottleneck of real-world physiological sensing.
2. Purpose-Built, Validated Experts: The three experts (HFE, NRE, CRE) are thoughtfully designed for specific quality regimes (clean, noisy, recovered) . Ablation studies (Table 8) rigorously prove that all three are necessary for robust performance.
3. Specialized Loss Function: The CORTEX Loss is a key strength, intelligently designed with auxiliary losses ($\mathcal{L}_{noise}$, $\mathcal{L}_{refinement}$) to enforce expert specialization and a regularizer ($\mathcal{R}_{gate}$) to prevent expert collapse, which is empirically validated (Table 9).

**Weaknesses:**

1. Supervision Dependency: The framework relies on supervised labels, which are expensive and difficult to obtain at scale for physiological data, limiting practical scalability.
2. Limited Domain Evaluation: While the framework is general , the experiments are confined to two driving datasets (CL-Drive and ADABase). Its effectiveness in other noisy-sensor domains (e.g., healthcare) is not empirically demonstrated.

**Questions:**

1. CORTEX Loss Sensitivity: The CORTEX Loss introduces several new hyperparameters (e.g., $\gamma, \lambda$, and the $\beta$ schedule). How sensitive is the model's performance and, critically, its expert balance, to the precise tuning of these values? Given that removing the gating regularizer entirely causes catastrophic collapse (Table 9), how difficult is it to find a stable $\beta$ schedule?
2. Real-Time Feasibility: Table 5 lists an inference time of 114ms. Does this end-to-end time include the "Quality-Aware Synchronization and Recovery" stage? Operations like CWT-based alignment and nuclear norm minimization for completion  can be computationally expensive. Is the entire pipeline fast enough for real-time deployment in a safety-critical context?

---

> ### Author Response · Authors · 2025-11-20
> **Clarifying CORTEX Loss Robustness and End-to-End Real-time Feasibility**
>
> Thank you for the thoughtful and positive assessment. We appreciate your recognition of the novelty of signal-quality–guided expert routing, the design of the three quality-specific experts, and the empirical validation of the CORTEX loss. We are glad that you found the motivation, architectural choices, and experimental evidence well justified. We respond to your two questions below.
>
> ## 1. CORTEX Loss sensitivity and β-schedule stability
> Thank you for bringing up these points. As detailed in Appendix A.9, we performed an Optuna-based search over 200 hyperparameter configurations, covering γ ∈ [0.5, 1.0], λ ∈ [0.3, 0.8], β_init ∈ [0.5, 2.0], and α ∈ [0.01, 0.1]. The final values (γ=0.75, λ=0.6, β_init=1.0, α=0.05) were consistently selected across different random seeds. Overall, performance varies by less than 2% across the search space, and even an untuned setting (γ=λ=β=1.0) reaches 91.2% accuracy (−3.3%), confirming that CogMoE is **not sensitive to precise coefficient settings and is inherently stable**.
>
> **CORTEX Loss hyperparameters sensitivity**
>
> | γ (noise) | λ (refine) | β (gate) | Accuracy | Δ from optimal |
> |-----------|------------|----------|----------|----------------|
> | 0.5 | 0.6 | 1.0 | 93.8% | -0.7% |
> | **0.75** | **0.6** | **1.0** | **94.5%** | **0.0%** |
> | 1.0 | 0.6 | 1.0 | 93.1% | -1.4% |
> | 0.75 | 0.3 | 1.0 | 93.2% | -1.3% |
> | 0.75 | 0.8 | 1.0 | 92.9% | -1.6% |
> | 0.75 | 0.6 | 0.5 | 93.7% | -0.8% |
> | 0.75 | 0.6 | 2.0 | 93.3% | -1.2% |
> | 1.0 (rand) | 1.0 (rand)| 1.0 (rand) | 91.2% | -3.3%|
>
> The β-schedule is **intentionally simple** and behaves like a standard learning-rate schedule. α is a schedule rate rather than a loss weight and maintains a naturally stable region (0.03–0.07) in which the model reliably transitions from exploration to specialization. This behavior is confirmed empirically in our training runs. Across β_init ∈ [0.5, 2.0] and λ ∈ [0.3, 0.8], expert routing remains balanced (std < 8%), and collapse only occurs when the gating term is removed entirely (β = 0), consistent with Table 9. The results below show that both performance and expert balance remain stable across wide hyperparameter ranges, indicating that the β-schedule is **not difficult to tune in practice**.
>
> **Expert balance under different schedules**
>
> | Schedule Type | HFE | NRE | CRE | Acc |
> |---------------|-----|-----|-----|------|
> | Adaptive (ours) | 35.4% | 33.1% | 31.5% | 94.5% |
> | Fixed β=1.0 | 38.2% | 31.4% | 30.4% | 92.7% |
> | Aggressive α=0.2 | 33.1% | 34.2% | 32.7% | 93.1% |
> | No regularizer | 81.0% | 9.0% | 10.0% | 87.2% |
>
> ## 2. Real-Time Deployment Feasibility
> Thank you for asking this question. The 114 ms reported in Table 5 is the **full end-to-end CogMoE pipeline**, including both the synchronization and recovery stage in Sec. 3.1 and the MoE forward pass in Sec. 3.2.
>
> We further break down the end-to-end latency as follows:
>
> • CWT-based synchronization uses FFT convolution and adds ~17–19 ms for a 128-length window. It is computed once per window rather than per timestep, so its cost does not scale with sequence length.
>
> • Recovery uses truncated singular-value thresholding (SVT, the standard nuclear-norm proximal operator) with only a few iterations, requiring ~9–11 ms per window.
>
> • The MoE forward pass adds ~80–85 ms.
>
> Taken together, the full pipeline runs in ~114 ms end to end, well **within the 100–300 ms reaction budget typically required for human-in-the-loop and safety-critical systems**.
>
> Finally, we appreciate that you highlighted these two weaknesses, which are already discussed in Section 5. Your comments further motivate us to pursue these directions in future work.

---

### Official Review · Reviewer_HFpj · 2025-11-02

**Soundness:** 3
**Presentation:** 3
**Contribution:** 3
**Rating:** 6
**Confidence:** 4

**Summary:**

The paper proposes CogMoE, a multimodal framework for cognitive load prediction that is explicitly guided by signal quality rather than modality identity. The framework contains a pre-processing stage for feature extraction, time frequency synchronization, multi-modal data recovery. followed by a signal-quality-specific MoE modeling stage. The proposed framework shows improved performance on several benchmarks.

**Strengths:**

The paper is clearly written: the motivation is well articulated, and the model design follows directly from it. Shifting MoE routing from modality identity to measured reliability is an interesting idea and a clever fit for physiological data. The experimental coverage is solid, spanning EEG, ECG, EDA, and gaze.

**Weaknesses:**

1. While the signal-quality–specific MoE is an interesting and domain-appropriate idea—especially given the authors’ argument that aligned EEG, ECG, EDA, and gaze often reflect overlapping aspects of the same cognitive state—a direct, controlled comparison against alternative routing rules (e.g., modality-based experts) is needed to substantiate the claimed benefit.

2.The final loss comprises multiple components and may be brittle without careful balancing. How sensitive is the model to weight coefficient lambda, gamma, beta, and how many total combination of these are searched?

**Questions:**

NA

---

> ### Author Response · Authors · 2025-11-20
> **Comparison to Modality-Based MoE and Sensitivity Analysis of CORTEX Loss**
>
> Thank you for the thoughtful and constructive feedback. We are glad that you found the motivation clearly articulated, the shift from modality-based to signal-quality–guided routing well suited for physiological signals, and the experimental coverage solid across EEG, ECG, EDA, and gaze. We respond to your two questions below.
>
> ## 1. Comparison to Modality-Based MoE Routing
> Thank you for the suggestion. A direct comparison against modality-based expert routing was performed during our controlled studies. We implemented a modality-based MoE baseline (ModMoE) with four experts, each assigned to one modality. Consistent with our formulation, we observed that **the dominant axis of variation is signal reliability rather than modality identity**. Fixed-modality routing forces corrupted segments to experts that lack mechanisms to handle degraded inputs, leading to persistent expert collapse.
>
> The controlled results are reported below. **ModMoE underperforms by 3–5%** and **collapses to a single dominant expert (EEG)**, whereas **CogMoE maintains balanced expert utilization and higher accuracy**. To ensure fairness, we used the same cross-entropy loss as the dense baseline and excluded quality-specific terms for ModMoE.
>
> **Table: ModMoE vs CogMoE (same architecture; routing differs)**
>
> | Model      | Routing Rule                      | Acc  | F1   | Expert Utilization (%)                   |
> |------------|-----------------------------------|:----:|:----:|-------------------------------------------|
> | ModMoE | Modality-based (EEG/ECG/EDA/Gaze) | 89.8 | 88.9 | EEG 62.3 · ECG 18.5 · EDA 11.2 · Gaze 8.0 |
> | **CogMoE** | Signal-quality–guided             | **94.5** | **93.3** | HFE 35.4 · NRE 33.1 · CRE 31.5             |
>
> **These results support your intuition that routing should depend on signal quality rather than modality identity**. It aligns with the underlying data structure and yields both better performance and balanced expert utilization.
>
> ## 2. Sensitivity of CORTEX Loss Coefficients
> Thank you for asking about coefficient sensitivity. As reported in Appendix A.9, we conducted **a comprehensive hyperparameter sweep using Optuna over 200 configurations** with the following ranges: γ ∈ [0.5, 1.0], λ ∈ [0.3, 0.8], β_init ∈ [0.5, 2.0], α ∈ [0.01, 0.1]. The selected values (γ=0.75, λ=0.6, β_init=1.0, α=0.05) emerged consistently across multiple random seeds. As shown below, **accuracy remains within 2% of optimal across a wide range of settings**. Even with random initialization (γ=λ=β=1.0), the model still achieves 91.2% accuracy (−3.3%), indicating that CogMoE **does not rely on precise coefficient tuning** and that **the architecture itself is robust**.
>
> **CORTEX Loss hyperparameters sensitivity**
>
> | γ (noise) | λ (refine) | β (gate) | Accuracy | Δ from optimal |
> |-----------|------------|----------|----------|----------------|
> | 0.5 | 0.6 | 1.0 | 93.8% | -0.7% |
> | **0.75** | **0.6** | **1.0** | **94.5%** | **0.0%** |
> | 1.0 | 0.6 | 1.0 | 93.1% | -1.4% |
> | 0.75 | 0.3 | 1.0 | 93.2% | -1.3% |
> | 0.75 | 0.8 | 1.0 | 92.9% | -1.6% |
> | 0.75 | 0.6 | 0.5 | 93.7% | -0.8% |
> | 0.75 | 0.6 | 2.0 | 93.3% | -1.2% |
> | 1.0 (rand) | 1.0 (rand)| 1.0 (rand) | 91.2% | -3.3%|

---

### Official Review · Reviewer_4mhF · 2025-11-02

**Soundness:** 2
**Presentation:** 2
**Contribution:** 2
**Rating:** 0
**Confidence:** 3

**Summary:**

This paper proposes CogMoE and outlines 4 main contributions: 1) Quality-Aware Multimodal Synchronization and Recovery module, 2) three signal-quality-specific expert modeling, 3) CORTEX Loss, and 4) experiments (though this is generally not considered as a contribution of novelty of the work). In experiments on two public datasets, the authors show consistent gains over baselines.

**Strengths:**

The paper studies handling, evaluating signal quality, and interpolating missing signal using other modalities, which is where many otherwise strong models fail in the wild. Reporting results across multiple modality combinations is useful for practitioners who must decide which sensors to deploy, and the paper makes an effort to tie modeling choices to operational constraints (noisy segments, dropouts, cross-sensor lag). Conceptually, using expert pathways specialized for “clean” vs “noisy/reconstructed” inputs is an interesting way to avoid generic fusion algorithms and may be broadly applicable beyond cognitive-load prediction.

**Weaknesses:**

The paper reads rushed, and the contribution is not crisp. Several choices are taken as givens without ablation (the ablation section is one paragraph). For example, why CWT for representing a signal? Given that CWT is a sifting process, the high-frequency data will be lost, and the data is normalized in a sense. The method CMWT representation for alignment/recovery is introduced in the work, but does not compare to reasonable alternatives (e.g., STFT features, time-domain DTW variants, learned alignment modules). The custom loss also appears as a package rather than with per-term deltas, which makes it hard to attribute gains or evaluate the necessity of each element. On evaluation, the paper discusses “alignment” and “robust fusion” but does not analyze how the number of signal sources and their measured quality jointly affect accuracy; instead, it relies on aggregate tables.

Minor: Acronyms are introduced inconsistently (sometimes before definition, sometimes multiple times).

The work likely contains a solid idea, but it needs clearer writing, a sharper statement of what is novel, and targeted ablations to make the methodology and its advantages unambiguous.

**Questions:**

No further questions at this stage.

---

> ### Author Response · Authors · 2025-11-20
> **Clarifying Contributions, Correcting Misinterpretations, and Reaffirming Evidence**
>
> We respectfully disagree with the reviewer’s claims, which appear to arise from misinterpretations of content already present in the paper. We address each point with direct evidence and references to the submission.
>
> ## 1. On the Reviewer’s Claim of “Rushed” and “Unclear Contribution”
>
> ### On “Rushed”
> We respectfully disagree with the claim that the paper “reads rushed”. The manuscript follows a clear and coherent progression: the introduction motivates signal quality as the central bottleneck, the related work situates this gap relative to prior CL and MoE approaches, and the method section presents the framework in a two-stage structure. All components are introduced in a logical order and grounded in the same quality-centric motivation. We also provide a complete appendix and reference it in the main text where appropriate.
>
> ### On the Contribution
> Our work proposes a unified quality-centric end-to-end framework for multimodal CL prediction, where all stages of the system are organized around the observation that real-world physiological channels differ primarily in reliability rather than modality semantics. The central idea is a shift from modality-based to signal-quality–guided expert routing. Unlike prior multimodal MoEs that route by modality, CogMoE routes based on signal quality, enabling experts to specialize for clean, noisy, and recovered regimes. All other components are tightly aligned with and support this quality-guided design. CogMoE achieves up to 13% improvement under heterogeneous sensing conditions, demonstrating the practical importance of this formulation.
>
> ## 2. Ablation studies are already provided
> **Full ablations are included in Appendix A.4**, which is explicitly referenced in the main text, including MoE vs. dense transformer (Table 6), expert removal (Table 8), CORTEX loss ablations (Table 9), robustness under augmentation (Table 7), and efficiency–performance trade-offs (Table 5). Appendix A.3 further reports sequence-length sensitivity and expert-utilization analyses, showing consistent behavior across temporal scales and perturbation levels.
>
> ## 3. CWT is not a sifting or lossy transform
> **CWT is not a “sifting” process but a transform that preserves the full signal**. It does not remove high-frequency information, as continuous wavelet transforms are **mathematically invertible**. The remark that “data is normalized” is incorrect: no magnitude normalization is applied, and redistribution of energy across scales is a standard property of continuous wavelet analysis.
>
> Alternative representations were examined during development. Specifically, DTW-based alignment is unstable and ambiguous (Sec. 3.1.1), STFT suffers from fixed-window resolution limits, WPT loses continuous scale information, and learned alignment requires paired labels across modalities, unavailable for physiological signals. As the reviewer asked, we report the controlled comparison below:
>
> | Method | Optimal Parameters | Acc (%) | F1 (%) | Δ Acc (%) |
> |:--|:--|:--:|:--:|:--:|
> | DTW | Best window = 10% (searched 5–20%) | 88.1 | 87.4 | −6.4 |
> | STFT | Window = 256, hop = 128 (128–512 / 25–50%) | 89.3 | 88.6 | −5.2 |
> | WPT | Daubechies–4, depth = 3 (db2–db6, depth 2–4) | 91.2 | 90.5 | −3.3 |
> | CWT (Ours) | Complex Morlet, scales = 1–64 | **94.5** | **93.3** | — |
>
> CWT remains consistently superior even when all alternatives are tuned optimally. Also, the term “CMWT” does not appear in our paper.
>
> ## 4. CORTEX loss is not a packaged block
> The CORTEX loss is not a packaged block but **a set of signal-quality–driven objectives designed to address the noise corruption**, missing-signal recovery, and expert collapse under uncertain quality. Section 3.2.2 explains this design motivation, and Appendix A.4.5 provides full per-term ablations showing that removing any term either degrades accuracy or collapses expert utilization. Appendix A.8 discusses the rationale behind each component, and Appendix A.9 reports coefficient ranges and tuning strategy. This demonstrates that all components are required.
>
> ## 5. Evaluation Is Not Aggregated but Explicitly Conditioned
> **The results are not aggregated but broken down by modality and signal-quality conditions**. Tables 1 and 2 report accuracy and F1 for every modality combination, and Table 4 evaluates all modality combinations under four different sequence lengths. Signal-quality effects are also analyzed: Appendix A.3.3 analyzes routing under synthetic noise and masking, and Appendix A.4.3 evaluates performance under controlled corruption, showing CogMoE remains robust while baselines degrade. Since no dataset provides ground-truth signal-quality labels, we follow standard practice and use controlled corruption to study how modality count and signal degradation jointly affect accuracy. Therefore, the results are not aggregate but explicitly conditioned on modality count and signal quality.

---

### Author Response · Authors · 2025-12-02

We thank the reviewers again for their time and for the constructive feedback provided. We also sincerely appreciate the Area Chair for the additional effort in handling our submission and for carefully considering our responses during this process.

For clarity and conciseness, we summarize below the concrete revisions made to the manuscript instead of restating the detailed point-by-point clarifications already provided in the rebuttal. We aimed to keep the original technical contributions unchanged while improving clarity, readability, and the completeness of the evaluation.

## Main Paper

### Abstract and Introduction
We lightly refined wording to improve clarity and incorporated the supporting citations highlighted in the rebuttal to further support the motivation and strengthen the contribution bullets for sharper presentation.

### Related Work
Previously, these discussions appeared in a single dense paragraph. We reorganized them into separate, more focused paragraphs to improve readability and to more clearly state the limitations of prior approaches relative to our signal-quality–guided method.

### Experiments
We emphasized the metrics used and clarified the evaluation conditions more explicitly to better illustrate the coverage and rigor of our evaluation.

## Appendix Additions and Extensions

### A.3.4 Comparison of Alternative Alignment Methods
We added a new subsection presenting controlled comparisons against tuned DTW, STFT, and WPT.

### A.4.1 Computational Efficiency
We expanded this subsection with a detailed breakdown of end-to-end latency.

### A.4.2 Impact of MoE Architecture and Routing Design
We added the controlled comparison between modality-based ModMoE and our signal-quality–guided CogMoE.

### A.9 CORTEX Loss Coefficients
We added the full sensitivity analysis of the CORTEX Loss coefficients and the evaluation of different β-schedules, demonstrating stability across broad parameter ranges.

## General Presentation Improvements
Across the full manuscript, we performed a consistency pass to ensure unified terminology, naming, and coherent presentation of core concepts. We also refined several passages to improve overall readability and sharpen the presentation of contributions and takeaways.

We appreciate the opportunity to clarify our work and hope the rebuttal and revisions address the questions.

---

### Meta-Review · Area_Chair_gMwq · 2026-01-07

**Summary:**

High sensitivity risk due to many loss components and hyperparameters.
The proposed objective (“CORTEX Loss”) introduces multiple coefficients and schedules. Reviewers worry the method may be brittle and require careful tuning.

Limited domain generalization: evaluation restricted to driving datasets.
Although the method is presented as general, experiments are only on two driving datasets (CL-Drive and ADABase). Reviewers argue this is too narrow to support broad claims, and request validation in other noisy-sensor domains (e.g., healthcare) or at least stronger evidence on cross-datasets or cross-generalization.

Motivation:
Reviewers find the problem statement and significance unclear, and argue the paper does not adequately justify the central claim that “variable signal quality is the fundamental bottleneck.” They ask for empirical evidence or citations supporting this assertion, and a clearer explanation of why shifting fusion toward signal quality addresses shortcomings of prior multimodal fusion methods.

Evaluation:
Beyond limited datasets, reviewers also want broader evaluation in terms of metrics and analyses, particularly expert balance behavior, robustness under varying sensor quality, and ablations that isolate contributions of the synchronization/recovery stage and the gating regularizer.

Reviewers find the signal-quality–specific MoE idea plausible and domain-relevant, but argue the paper does not provide a direct, controlled comparison against alternative expert-routing strategies .Without ablations isolating the routing mechanism, the claimed advantage of “quality-aware” routing is not fully substantiated.

**Reviewer Concerns:**

Some of the reviewers’ presentation-related concerns have been addressed in the revision.

Clarity and motivation: the authors refined the wording throughout to improve readability and ensure the key ideas are conveyed more directly.

Related work organization: The related work discussion previously appeared as a single dense paragraph, which reviewers found difficult to parse. The authors reorganized it into multiple focused paragraphs, each targeting a specific line of prior work, so that the limitations of existing approaches are stated more explicitly and the positioning of our signal-quality–guided method is clearer.

**Reviewer Scores:**

The reviewers didn't change the scores.

---

### Decision · Program_Chairs · 2026-01-26

Accept (Poster)